# Lipase-mediated detoxification of host-derived antimicrobial fatty acids by *Staphylococcus aureus*
Arnaud Kengmo Tchoupa [1,2,3] ✉, Ahmed M. A. Elsherbini [1,2,3,6], Justine Camus [1,2,3,6], Xiaoqing Fu[4], Xuanheng Hu[1,2,3], Oumayma Ghaneme[1,2,3], Lea Seibert[1,2,3], Marco Lebtig[1,2,3], Marieke A. Böcker[1,2,3], Anima Horlbeck[1,2,3], Stilianos P. Lambidis[1,2,3], Birgit Schittek[2,5], Dorothee Kretschmer [1,2,3], Michael Lämmerhofer [4] & Andreas Peschel [1,2,3]

Long-chain fatty acids with antimicrobial properties are abundant on the skin and mucosal surfaces, where they are essential to restrict the proliferation of opportunistic pathogens such as *Staphylococcus aureus*. These antimicrobial fatty acids (AFAs) elicit bacterial adaptation strategies, which have yet to be fully elucidated. Characterizing the pervasive mechanisms used by *S. aureus* to resist AFAs could open new avenues to prevent pathogen colonization. Here, we identify the *S. aureus* lipase Lip2 as a novel resistance factor against AFAs. Lip2 detoxifies AFAs via esterification with cholesterol. This is reminiscent of the activity of the fatty acid-modifying enzyme (FAME), whose identity has remained elusive for over three decades. In vitro, Lip2-dependent AFA-detoxification was apparent during planktonic growth and biofilm formation. Our genomic analysis revealed that prophage-mediated inactivation of Lip2 was rare in blood, nose, and skin strains, suggesting a particularly important role of Lip2 for host – microbe interactions. In a mouse model of *S. aureus* skin colonization, bacteria were protected from sapienic acid (a human-specific AFA) in a cholesterol- and lipase-dependent manner. These results suggest Lip2 is the long-sought FAME that exquisitely manipulates environmental lipids to promote bacterial growth in otherwise inhospitable niches.

At the host-pathogen interface, lipids exert multifaceted functions as, for instance, building blocks for cells and extracellular matrices[1–3], energy sources[4,5], entry routes into host cells[6], immunomodulators[7], and potent antimicrobials[8–10]. To harness environmental lipids and fuel their growth, bacteria utilize a plethora of lipolytic enzymes, whose substrates include sphingolipids, phospholipids, and triacylglycerols[4,11–13]. These lipid hydrolases release host-derived long-chain fatty acids with antibacterial properties, also referred to as antimicrobial fatty acids (AFAs)[14]. An intriguing concept is that bacteria would secrete lipases to release AFAs from complex lipids and thereby inhibit AFA-susceptible competitors within the same niche, for instance on human skin. This has been demonstrated for *Corynebacterium accolens* and *Streptococcus pneumoniae*[12]. Hence, adaptation strategies to AFAs represent a prerequisite for stable colonization of the skin

and mucosal surfaces. *Staphylococcus aureus*, an opportunistic pathogen colonizing asymptomatically the nares of ~30% of the human population[15], is no exception.

The intermittent skin colonization by *S. aureus* in healthy individuals (10–20%) clearly contrasts with the nearly persistent colonization of patients with dermo-inflammatory disorders like atopic dermatitis (80–100%)[16]. Interestingly, atopic dermatitis has been associated with several lipid disorders, including defects in sapienic acid, a potent human-specific AFA[17]. It is unclear whether *S. aureus* strains associated with atopic dermatitis are exceptionally impervious to AFAs. The diverse resistance mechanisms used by *S. aureus* against AFAs have been reviewed elsewhere[14]. Notably, the bacterium has long been known to secrete a fatty acid-modifying enzyme (FAME) that mediates AFA-detoxification via esterification with cholesterol

[1]Interfaculty Institute of Microbiology and Infection Medicine Tübingen, Infection Biology Section, University of Tübingen, Tübingen, Germany. [2]Cluster of Excellence EXC 2124 Controlling Microbes to Fight Infections, University of Tübingen, Tübingen, Germany. [3]German Center for Infection Research (DZIF), Partner Site Tübingen, Tübingen, Germany. [4]Institute of Pharmaceutical Sciences, University of Tübingen, Tübingen, Germany. [5]Dermatology Department, University Hospital Tübingen, Tübingen, Germany. [6]These authors contributed equally: Ahmed M. A. Elsherbini, Justine Camus. ✉ e-mail: arnaud.kengmo-tchoupa@uni-tuebingen.de

or, with lower efficacy, other alcohols[18]. The identity of the protein(s) responsible for FAME activity has remained elusive.

To uncover FAME and other protective strategies against the deleterious effects of AFAs, proteins secreted by *S. aureus* grown in the presence of a subinhibitory concentration of AFAs have been examined[19]. This study revealed that the bacterium boosted its release of the lipolytic lipase Lip2 (also referred to as Geh or Sal2) when primed with AFAs[19]. Recently, we uncovered Lip2 and other lipases as major components of membrane vesicles (MVs) from *S. aureus* irrespective of the presence of AFAs in the growth medium[20]. Given that the impact of *S. aureus* lipases on bacterial susceptibility to AFAs in various lipid environments has never been thoroughly investigated, the protective effects of lipase-loaded MVs against AFAs[20] prompted us to probe the role of lipases in bacterial adaptation to AFAs.

Here, we unveiled Lip2 as an unanticipated resistance factor against AFAs. Lip2 is necessary and sufficient for the esterification of AFAs to cholesterol, with consequences for bacterial growth in liquid cultures, biofilms, and on mammalian skin.

## Results

### *S. aureus* lipases mediate resistance against AFAs

Our recent proteomics study has uncovered lipases as major components of MVs from *S. aureus* even when the bacterium was grown in the presence of AFAs[20]. These observations suggest that bacteria utilize lipases to cope with AFAs. In agreement with the previously reported protective roles of MVs against AFAs[20], we hypothesized that lipases are required for bacterial growth in the presence of AFAs. To test this hypothesis, we monitored the growth kinetics of wild-type USA300 JE2 (WT) or its mutant[21] defective for both Lip1 and Lip2 lipase production (henceforth referred to as Δlip) in a rich medium where Δlip displayed no growth defect (Fig. 1a, b). Notably, even upon treatment with palmitoleic acid (PA), a major AFA of mammalian skin[22] and nasal fluid[9], no clear differences in growth behaviors were

apparent between Δlip and WT, which were both strongly inhibited by 50 μM PA, i.e., PA concentration in the nasal fluid[9] (Supplementary Fig. 1 and Fig. 1a, b). The abundance of PA generally correlates with that of cholesterol in the nasal fluid[9]. Owing to cholesterol-protective roles against AFAs[14], we wondered whether cholesterol would boost the growth of WT and Δlip in the presence of otherwise inhibitory amounts of PA. Strikingly, cholesterol, which alone does not alter the replication of *S. aureus* (Supplementary Fig. 1), counteracted PA toxicity in a lipase-dependent manner (Fig. 1a, b). In addition to optical density readings, the lipase-dependent protective effects of cholesterol were also evidenced by CFU (colony forming unit) enumeration upon bacterial growth in a rich medium (Fig. 1c) or in a chemically defined medium (Supplementary Fig. 1). Since triacylglycerols are important lipids of human sebum and skin surface, which bacterial lipases use as substrates to release AFAs[12,23], we ruled out the possibility that the presence of glycerol acyl tripalmitoleate could affect cholesterol-dependent AFA resistance (Supplementary Fig. 1).

Linoleic acid (LA) is an AFA found virtually everywhere throughout the human body, including the nasal fluid[9]. In human stratum corneum LA concentration is estimated to be ~ 2 mM[24,25], while LA can reach up to 5 mM in the blood[26]. The heightened susceptibility of Δlip to AFAs was readily apparent when LA was used alone (Supplementary Fig. 2). In experimental settings where WT and Δlip were similarly inhibited by LA, cholesterol was protective only for WT when bacteria grown to stationary phase were used as inoculum for growth assays (Supplementary Fig. 2). In contrast, when bacteria were grown to exponential phase prior to growth assays, WT and Δlip were equally susceptible to LA, but cholesterol-mediated protection against LA was residual (Supplementary Fig. 2). Strikingly, the growth of exponential phase bacteria was completely inhibited by 100 μM LA whereas 200 μM LA could not abrogate the growth of stationary phase WT (Supplementary Fig. 2). In addition to the observed increased susceptibility of exponential phase bacteria to LA, the production of lipases, not yet fully

**Fig. 1 | Lipases protect *S. aureus* against palmitoleic acid. a** Optical density at 600 nm (OD$_{600}$) was measured over 24 h to monitor the growth of USA300 JE2 (WT; black) and its Lip1- and Lip2-defective double mutant (Δlip) in plain nutrient broth (NB), or NB supplemented with 50 μM palmitoleic acid (PA) or 50 μM PA and 50 μM cholesterol (Chol). **b** Area under the growth curves (shown in **a**) was computed in arbitrary units (AU). **c** Viable WT and Δlip were enumerated upon growth for 24 h in NB, or NB supplemented with 50 μM PA or 50 μM PA + 50 μM Chol. **d** WT and Δlip bearing an empty plasmid (pEmpty), and Δlip complemented with p*lip2* were grown as described in (**c**) while OD$_{600}$ was measured. Data are presented as means ± standard error of the mean (SEM) for 3 (**c**) or 4 (**a**, **b**, **d**) biological replicates. Statistical significance was determined by one-way analysis of variance (ANOVA) with Tukey's multiple comparisons test. ***$p$ = 0.0002, ****$p$ < 0.0001.

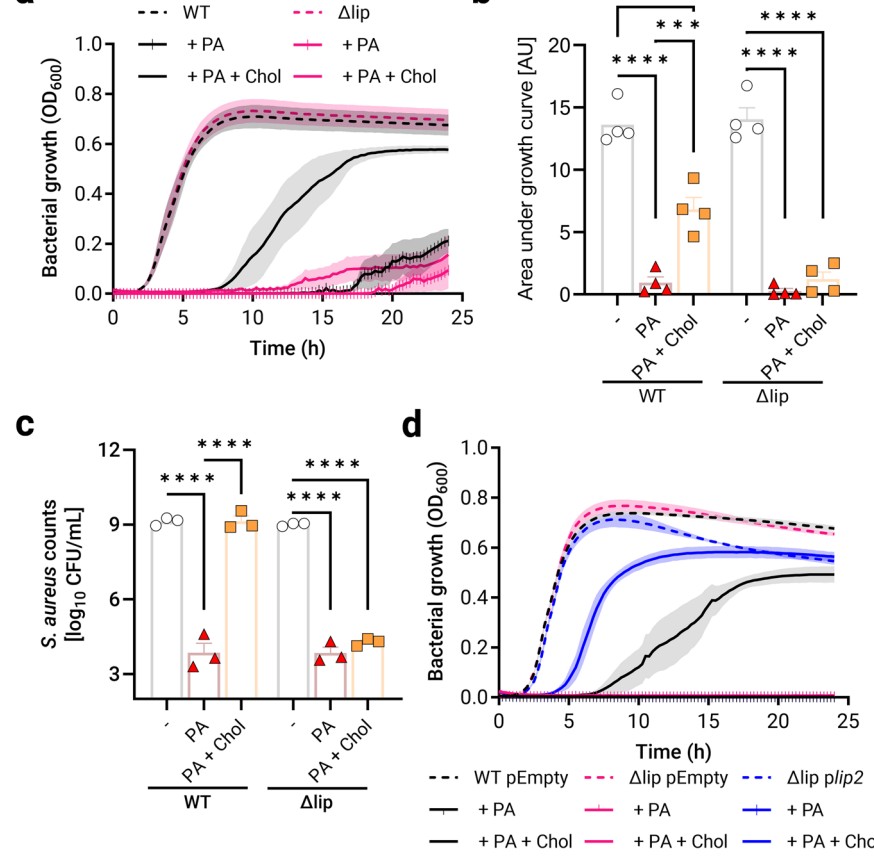

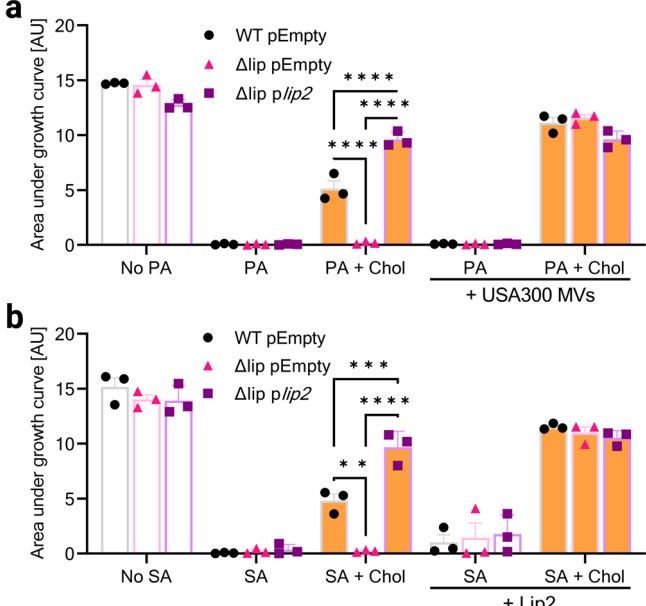

**Fig. 2 | Exogenous lipases enable cholesterol-dependent growth in the presence of AFAs. a** Wild-type USA300 JE2 and its isogenic Δlip mutant with pEmpty, and Δlip complemented with p*lip2* were grown in plain nutrient broth (NB), or NB with or without USA300 membrane vesicles (MVs) and supplemented with 50 μM palmitoleic acid (PA) or 50 μM PA + 50 μM cholesterol (Chol). Computed area under growth curves was plotted. **b** Area under the curves of the strains described in (**a**) upon growth in NB, or NB supplemented with 50 μM sapienic acid (SA) or 50 μM SA + 50 μM Chol, with or without recombinant Lip2. Data shown are means plus SEM ($n = 3$). Statistical significance was evaluated by two-way ANOVA with Tukey's multiple comparisons test. $**p = 0.0019$, $***p = 0.0009$, $****p < 0.0001$.

activated by the quorum-sensing system Agr[27] in these cells, may explain why cholesterol failed to protect these bacteria from LA. Therefore, all the follow up experiments were conducted with stationary phase bacteria. Taken together, the data strongly suggest that lipases Lip1 and/or Lip2 are necessary for *S. aureus* to benefit from cholesterol against AFAs.

## The lipase Lip2 is sufficient for cholesterol-mediated protection against AFAs

To determine whether Lip1 and Lip2 were both required for the phenotype of the double lipase mutant Δlip or one of both enzymes played a dominant role, we first tested a single *lip2* mutant[23] (Δ*lip2*) and its otherwise isogenic USA300 wild-type strain for growth in the presence of LA. Δ*lip2* displayed a longer lag phase (~11 h) compared to its WT (~7 h), suggesting that Lip2 is protective against LA (Supplementary Fig. 2). Next, Δlip was complemented with *lip2* on a plasmid (p*lip2*). The complemented strain Δlip p*lip2* had no growth advantage in rich medium over a Δlip mutant carrying an empty plasmid (pEmpty). However, p*lip2*-complementation enabled Δlip to proliferate in the presence of toxic amounts of PA (Fig. 1d), LA or sapienic acid (SA) (Supplementary Fig. 3), albeit only upon addition of cholesterol. The growth defect of Δlip pEmpty in media supplemented with cholesterol and AFAs, as compared to either WT pEmpty or Δlip p*lip2*, was alleviated when this mutant was provided with MVs from WT USA300 (Fig. 2a). MV-associated lipases appeared to be responsible for MV-mediated complementation of Δlip pEmpty since the AFA-resistance was not restored when MVs were derived from Δlip (Supplementary Fig. 3). Importantly, recombinant Lip2 also enabled the growth of Δlip pEmpty upon exposure to AFAs and cholesterol (Fig. 2b).

## Catalytically active Lip2 is required for cholesterol-mediated resistance to AFAs

In addition to the prominent role of Lip2 in cholesterol-mediated protection against AFAs, we sought to investigate a possible involvement of

Lip1. Therefore, Δlip was complemented with p*lip1*. The generated strain was then tested along with pEmpty-bearing WT and Δlip, as well as p*lip2*-complemented Δlip. In clear contrast to p*lip2*, p*lip1* did not allow Δlip to benefit from cholesterol and thereby grow in the presence of PA (Fig. 3a–d), suggesting that Lip2 is solely responsible for cholesterol-aided protection against AFAs. Next, to test whether the catalytic activity of Lip2 was required to mediate cholesterol-dependent AFA resistance, we genetically engineered p*lip2* into p*lip2*^S412A, bearing a catalytically inactive copy of Lip2 (Lip2 S412A), as demonstrated in previous studies[13,23]. Upon complementation with this catalytically inactive form of Lip2, the double mutant Δlip displayed no lipase activity, as assessed with a long-chain fatty acid ester substrate (Supplementary Fig. 4). This mutant was also unable to benefit from cholesterol supplementation to grow in the presence of PA (Fig. 3e), SA or the polyunsaturated α-linolenic acid (Supplementary Fig. 4). Taken together, our data indicate that Lip2 requires its enzymatic activity to mediate cholesterol-dependent AFA resistance.

## Cholesterol-mediated protection against AFAs is widespread in *S. aureus*

To investigate whether cholesterol was protective against AFAs for *S. aureus* strains other than USA300, USA400 MW2, USA200 UAMS-1, SH1000 and Newman were assessed for growth in the presence of cholesterol and PA. Except for Newman, all these *S. aureus* strains clearly benefited from cholesterol to better grow in the presence of PA (Fig. 4a). In agreement with our data, Newman Lip2-encoding gene (*lip2*) is disrupted by a prophage[28]. Upon complementation with p*lip2*, Newman also became able to replicate in a cholesterol-dependent manner at otherwise toxic concentrations of PA in planktonic conditions (Fig. 4b), or within biofilms (Fig. 4c).

## Lip2 mediates esterification of AFAs with cholesterol

*S. aureus* and many other staphylococci are known to utilize FAME to detoxify unsaturated, antimicrobial fatty acids by esterification with hydroxylated substrates, including cholesterol[18,29]. The protein responsible for FAME activity has been enigmatic for three decades. In the light of the Lip2-dependent protective effects of cholesterol against AFAs, we sought to test recombinant Lip2[23] for FAME (esterification) activity, which would result in a gradual decrease in free cholesterol and fatty acids. Indeed, Lip2 induced a decline in free cholesterol within a few minutes of incubation with LA (Fig. 5a). We probed the reproducibility of our results with ten times less lipids. Even in these settings, Lip2 mediated the gradual decrease of free cholesterol in the presence of LA (Supplementary Fig. 5).

Next, to investigate whether free cholesterol was replaced by esterified cholesterol upon incubation of Lip2 with LA and cholesterol, we utilized high-performance thin layer chromatography (HPTLC). Via HPTLC, we detected cholesteryl linoleate, a cholesteryl ester (CE18:2; Fig. 5b) only when Lip2, cholesterol and LA were co-incubated. Importantly, increasing LA concentrations resulted in Lip2-mediated cholesteryl ester production in a dose-dependent manner (Supplementary Fig. 5), arguing for the substrate-specific catalytic activity of purified Lip2. Lip2-catalyzed esterification of LA with cholesterol was also confirmed via ultra-high-performance liquid chromatography-electrospray ionization-tandem mass spectrometry (UHPLC-MS/MS) (Fig. 5c). Lip2 but not catalytically inactive Lip2 S412A displayed esterifying activity on all five AFAs we tested, irrespective of chain length and degree of unsaturation, as revealed by HPTLC (Supplementary Fig. 5).

When we probed Lip2-mediated esterification of cholesterol with palmitoleic acid (PA) in the presence of glycerol acyl tripalmitoleate (GTP), we observed that GTP did not prevent the FAME (esterification) activity of Lip2 at pH 6 (optimum pH for FAME activity[30]). However, the FAME activity of Lip2 was residual at pH 8, the optimum pH for lipase activity[23] (Supplementary Fig. 5). Further, to demonstrate that Lip2 released by *S. aureus* can esterify cholesterol with AFAs, we treated *S. aureus*-conditioned media from plasmid-bearing Δlip or WT strains with AFAs and cholesterol prior to lipid analysis. Cholesteryl esters (CE) were detected by HPTLC

**Fig. 3 | Catalytically active Lip2 is required for cholesterol-mediated resistance to AFAs.** Wild-type USA300 JE2 and its isogenic Δlip mutant with pEmpty, and Δlip complemented with either p*lip1* or p*lip2* were grown for 24 h in (**a**) basic medium (BM), or BM supplemented with (**b**) 100 μM palmitoleic acid (PA), (**c**) 100 μM cholesterol (Chol), or (**d**) 100 μM PA + 100 μM Chol. Cultures were then serially diluted and spotted onto TSB plates. Dilution factors are indicated below spots. One representative biological replicate (n = 3) is shown. **e** Area under the curves of pEmpty-bearing wild-type USA300 JE2 and its Δlip mutant, and Δlip complemented with either p*lip2^S412A* or p*lip2* cultured for 24 h in BM, or BM supplemented with 100 μM PA or 100 μM PA + 100 μM Chol. Data shown are means plus SEM (n = 3 or 4). Statistical significance was evaluated by one-way ANOVA with Tukey's multiple comparisons test. ****$p < 0.0001$.

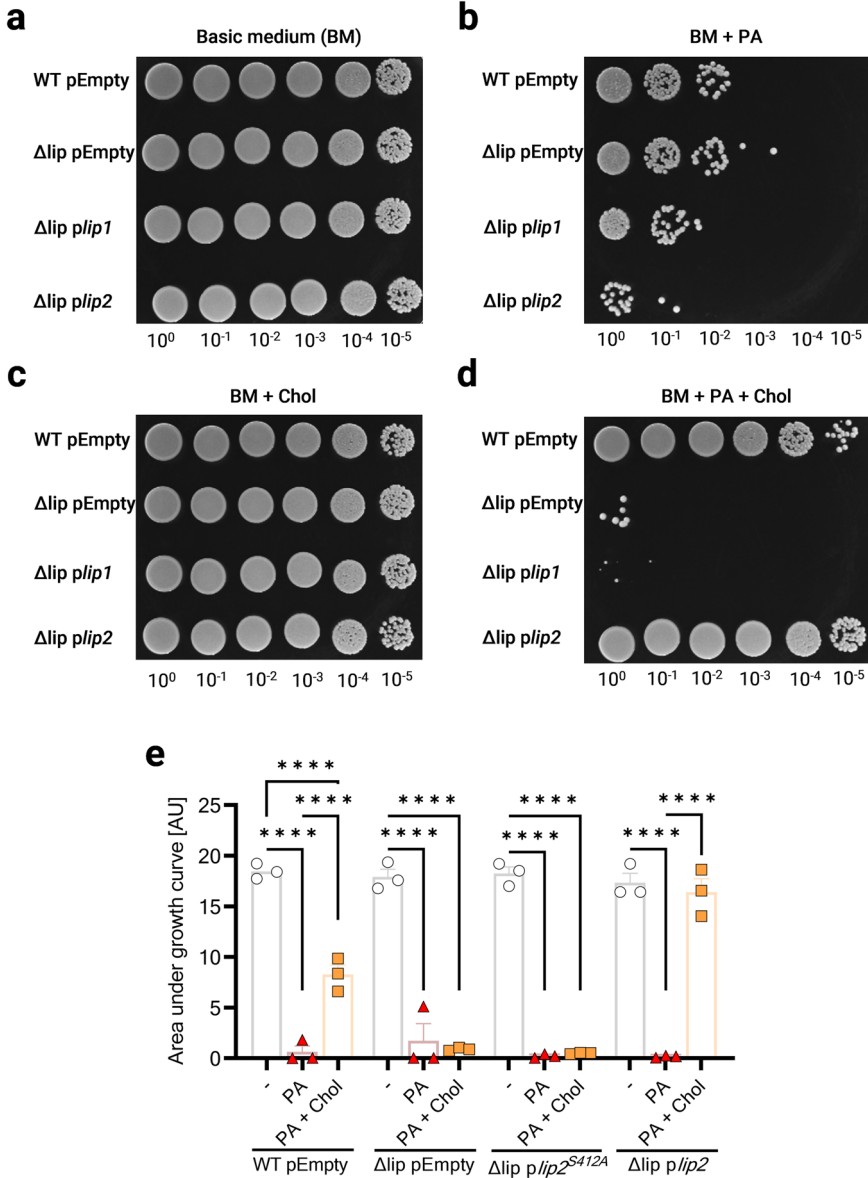

(Supplementary Fig. 6) or UHPLC-MS/MS (Fig. 5d) only in samples from Lip2-expressing strains, namely WT pEmpty and Δlip p*lip2*. CE production was concomitant with decreased concentrations of free AFA (Fig. 5e) and cholesterol (Supplementary Fig. 6). Taken together, our data identify Lip2 as FAME, which detoxifies AFAs by esterification with cholesterol.

Despite a clear preference for cholesterol, FAME has also been shown to use other alcohols for AFA esterification[18]. Consistent with Lip2-mediated FAME activity, CE were still produced by Lip2-expressing strains, when *S. aureus*-conditioned media were supplemented with approximately eight hundred times molar excess of ethanol to compete with cholesterol for AFA esterification (Supplementary Fig. 6). Moreover, this experimental setup unveiled that while Δlip p*lip2* or WT pEmpty esterified AFA with either ethanol or cholesterol, Δlip complemented with p*lip1* esterified AFA with ethanol only (Supplementary Fig. 6), suggesting FAME activity for Lip1 with ethanol and presumably other alcohols as substrates. The requirement of Lip2 for cholesterol esterification was also evidenced in the USA400 strain, MW2 WT, in which single mutants defective in either Lip1 (MW2 Δ*lip1*) or Lip2 (MW2 Δ*lip2*) were generated. Conditioned medium by MW2 Δ*lip2* could esterify ethanol but not cholesterol (Supplementary Fig. 6), while MW2 Δ*lip1*-conditioned medium retained the MW2 WT's ability to utilize

ethanol and cholesterol for AFA esterification (Supplementary Fig. 6). Together, these data underline cholesterol as preferred substrate for Lip2-mediated esterification of AFAs. As exemplified by Lip1, the ability to modify AFAs with alcohols is a poor predictor for cholesterol utilization and likely explains why FAME has remained elusive for so long.

## Cholesterol does not prevent membrane-damaging effects of AFAs

Our data (Fig. 5 and Supplementary Figs. 5 and 6) strongly suggest that the protective effects of cholesterol against AFAs are due to the Lip2-mediated esterification/detoxification of AFAs with cholesterol. However, additional, or alternative mechanisms might contribute to our observations. Recently, Lip2 was identified as one of the extracellular proteins binding to the cell surface of *S. aureus*[31]. We reasoned that surface-exposed Lip2 may mediate binding to cholesterol and lead to the formation of bacterial aggregates with decreased susceptibility to AFAs. To assess this possibility, we used dehydroergosterol (DHE) as a fluorescent cholesterol analog[32] for binding assays with pEmpty-bearing USA300 WT and Δlip, as well as Δlip complemented with p*lip1*, p*lip2* or p*lip2^S412A*. We did not observe any difference between Lip2-defective and Lip2-proficient strains in their ability to bind sterols (Fig. 6a). This suggests that impaired cholesterol

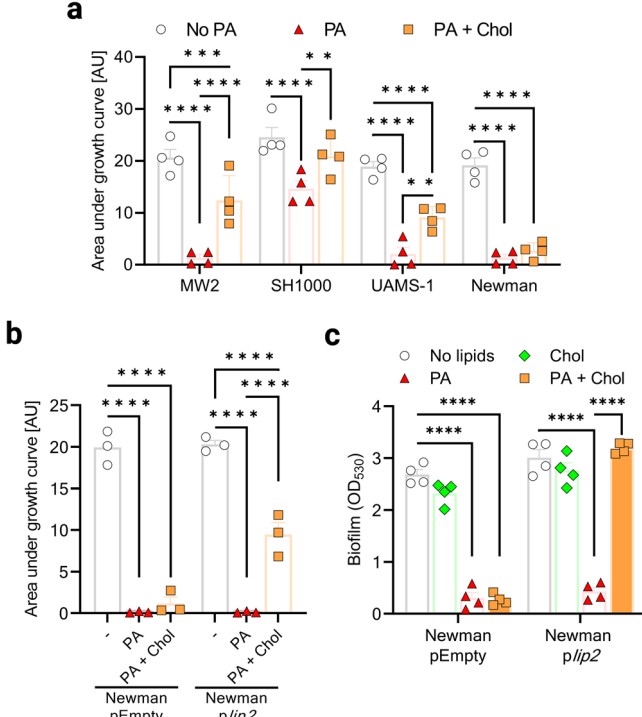

**Fig. 4 | Cholesterol-mediated protection against AFAs is widespread in *S. aureus*.**
**a** *S. aureus* strains (MW2, SH1000, UAMS-1, and Newman) were cultured for 24 h in basic medium (BM), or BM supplemented with 100 µM palmitoleic acid (PA) or 100 µM PA and 100 µM cholesterol (Chol). Growth was computed as area under the curves. **b** Area under the curves of Newman with either pEmpty or p*lip2* grown for 24 h in plain BM, or BM supplemented with 50 µM PA or 50 µM PA + Chol. **c** Optical density at 530 nm (OD$_{530}$) was measured after safranin staining of biofilms formed after 24 h by Newman with either pEmpty or p*lip2* in plain BM, or BM supplemented with 50 µM Chol, 50 µM PA, or 50 µM Chol + 50 µM PA. Data shown are means plus SEM for 3 (**b**) or 4 (**a**, **c**) biological replicates. Statistical significance by two-way ANOVA with Tukey's multiple comparisons test. *$p < 0.05$, **$p < 0.01$, ***$p < 0.001$, ****$p < 0.0001$.

binding is unlikely to be the reason why Lip2-deficient *S. aureus* failed to utilize cholesterol against AFAs.

Despite similar binding to sterols, it remained plausible that Lip2-deficient bacteria were defective in utilizing cholesterol to prevent early interactions with AFAs. We took advantage of a palmitoleic acid analog (PA alkyne) and click chemistry with azide fluor 488 for AFA-binding studies[20,33] with or without cholesterol supplementation. We found that, irrespective of Lip2 expression and despite cholesterol treatment, PA alkyne bound to WT and mutants, as revealed by flow cytometry upon 20-min incubation with PA alkyne (Fig. 6b). Thus, our results suggest that neither Lip2 nor cholesterol prevents the initial interaction of *S. aureus* membrane with AFAs.

Another putative protective mechanism of cholesterol could be to preserve the membrane integrity of Lip2-expressing bacteria in the presence of AFAs, which would be reminiscent of the role of the golden carotenoid pigments staphyloxanthin in *S. aureus*[34]. Since membrane-damaging effects of AFAs include loss of membrane potential[35], we examined the membrane potential of WT and mutants upon 30 min-treatment with PA, or PA and cholesterol. PA-treated bacteria displayed an almost undetectable membrane potential, which was not restored by co-treatment with cholesterol (Fig. 6c). This suggests that cholesterol per se does not prevent initial membrane damages caused by AFAs. Thus, Lip2-mediated esterification of AFAs with cholesterol seems to be the only mechanistic explanation for the protective effects of cholesterol toward AFAs. We reasoned that after the initial PA-provoked damage, gradual AFA detoxification would restore membrane potential. Therefore, we treated bacteria for 24 h with PA, or PA and cholesterol before membrane potential studies. Only Lip2-proficient strains

could recover from PA exposure in the presence of cholesterol and displayed a membrane potential similar to that of untreated controls (Fig. 6d).

## Lip2 is a conserved protein that can be disrupted by prophages
To gain unprecedented insights into a potential involvement of Lip2 into tissue tropism, we delved into our custom database of almost four thousand genomes of *S. aureus* obtained from the Bacterial and Viral Bioinformatics Resource Center (BV-BRC)[36] to determine whether the presence or absence of intact *lip2* is preferentially associated with specific *S. aureus* clones or specific human habitats. Our database encompasses blood (1481), nose (1587), and skin (767) isolates. An in silico polymerase chain reaction[37] was used to retrieve sequences of *lip2* in 91.23% (1352 out of 1481), 88.78% (1409 out of 1587), or 95.2% (730 out of 767) of blood, nose, or skin isolates, respectively (Fig. 7a). Next, Lip2 protein sequences were deduced from *lip2* genes. In keeping with the widespread presence of lipases in staphylococci[38], Lip2 appeared to be highly conserved in *S. aureus* strains irrespective of the isolation site (Supplementary Fig. 7). Interestingly, across the seven major sequence types (ST) of our database, the ST dictated Lip2 diversity (Supplementary Fig. 8). Irrespective of ST, eight mutation hotspots were apparent in Lip2 (Supplementary Fig. 9), with some mutations cooccurring in several clonal groups (Supplementary Table 1). It remains to be elucidated whether these modifications impact Lip2 lipase/FAME activity.

The nucleotide sequence of *lip2* encompasses a conserved integration site for prophages. A disruption of *lip2* gene by a prophage inactivates Lip2[13]. Therefore, we had a second look at *lip2* sequences to investigate how often prophage insertion occurred. Strikingly, only 2% (71 out of 3491) of the strains exhibited a prophage-disrupted *lip2*. Roughly half of the strains with prophage-disrupted *lip2* were from the sequence type ST398, a livestock-associated *S. aureus* lineage, which represents only 8% of the genomes in our database (Supplementary Fig. 9). Remarkably, prophage-mediated inactivation of *lip2* was more frequent in blood and nose isolates (2.1% and 2.7%, respectively) than in skin isolates (0.7%) (Fig. 7b). These results suggest that an intact *lip2* may be required for successful colonization and/or infection by *S. aureus*.

## Cholesterol contributes to the protection of *S. aureus* from AFAs on the skin
To ascertain the requirement of lipases for skin colonization in vivo, we opted for a well-established mouse skin colonization model[21,39], which mimics human atopic dermatitis. This model leverages the impaired skin barrier function upon extensive tape-stripping to improve skin colonization by *S. aureus* in a similar manner as in human atopic dermatitis patients. The tape-stripped skin was topically colonized with *S. aureus*. With such a model, we previously observed that wild-type USA300 JE2 and Δ*lip* did not differ in their capacity to colonize mouse skin[21]. Since tape-stripping is known to deplete lipids from the skin[40,41], we repleted mouse skin with sapienic acid (SA), or cholesterol plus SA during colonization with either WT or Δ*lip*. As bacteria were washed prior to in vivo experiments, we confirmed that cholesterol still protected washed bacteria from AFAs in vitro (Supplementary Fig. 10). In vivo, we observed that skin colonization by Δ*lip* was largely unaffected by cholesterol application, whereas Lip2-proficient WT appeared to benefit from cholesterol to better colonize the skin in the presence of SA (Fig. 7c, d). Taken together, our results suggest that *S. aureus* utilizes its lipases to manipulate environmental lipids and proliferate on the skin.

## Discussion
The immense success of *S. aureus* as an opportunistic pathogen requires strategies to circumvent host defenses, including AFAs[8,42]. The huge variety of the resistance mechanisms used by bacteria against AFAs strongly suggests a key role for AFAs at the host–pathogen interface[14]. Importantly, bacteria utilize a vast array of lipases to hydrolyze lipids in their environment with sometimes fatal consequences for microbial competitors[12,43] or eukaryotic host cells[44]. Bacteria-mediated lipid hydrolysis releases long-chain fatty acids, which can be toxic to microbes[12,23,45]. For *S. aureus* and other staphylococci, it is currently thought that lipase-expressing strains utilize FAME to detoxify

**Fig. 5 | Lip2 mediates esterification of AFAs with cholesterol. a** Upon mixing 300 μM cholesterol (Chol) with 300 μM linoleic acid (LA), free cholesterol was quantified before ($t = 0$) or following the addition of recombinant Lip2 (1.4 ng/ml) to the sample and incubation for the indicated duration. **b** HPTLC of a lipid standard mixture (Chol, LA, and cholesteryl ester CE18:2) and lipid extracts after 20-h incubation of Chol and/or LA with or without recombinant Lip2. **c** Structure of cholesteryl linoleate (CE 18:2) and representative extracted ion chromatograms of $m/z$ 648.585 ± 0.010 (precursor type $[M + NH4]^+$ in positive ion mode of CE 18:2) demonstrating detection of CE 18:2 upon co-incubation of LA, Chol and recombinant Lip2. **d, e** UHPLC-MS/MS lipid analysis upon 20-h incubation of *S. aureus*-conditioned media from the indicated strain (WT pEmpty, Δlip pEmpty, Δlip p*lip1*, Δlip p*lip2*, or Δlip p*lip2*$^{S412A}$) with Chol and LA. **d** CE 18:2 and (**e**) LA were measured. Bar graphs are means plus SEM for three (**a**) or five replicates (**d, e**). Statistical significance by one-way ANOVA with Dunnett's test relative to WT pEmpty. **p = 0.0034, ****p < 0.0001.

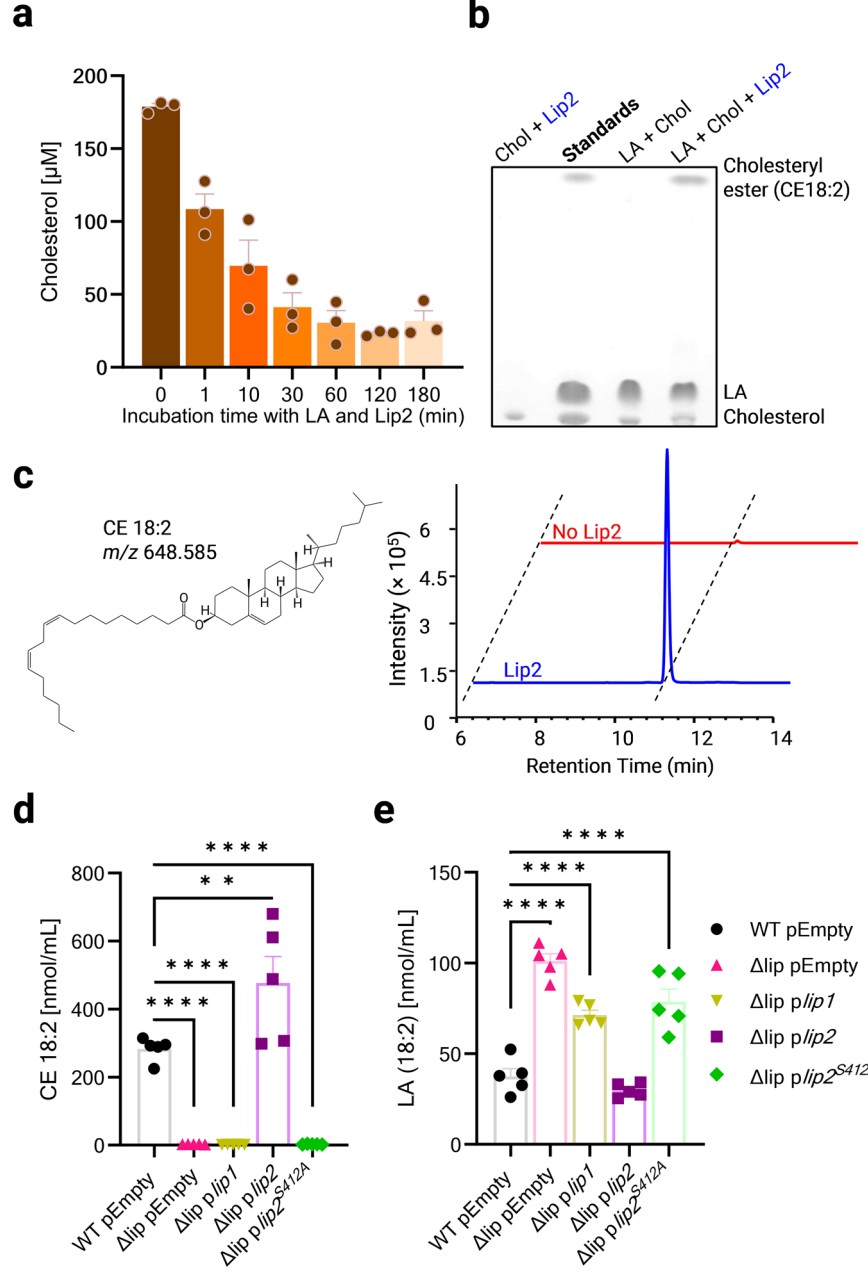

AFAs released by lipases. However, the identity of the protein(s) responsible for FAME activity has remained elusive for over three decades. Here, we uncovered that lipases are responsible for FAME activities in *S. aureus*. While both lipases Lip1 and Lip2 use ethanol and likely other alcohols for AFA esterification, only Lip2 esterifies AFAs with cholesterol. The ability to utilize cholesterol proved vital as cholesterol protected Lip2-proficient strains against AFA toxicity in planktonic as well as biofilm settings. The unanticipated substrate flexibility of Lip2 strongly suggests a more complex role for bacterial lipases in shaping the host lipid landscape than previously thought, with potential consequences for the microbiome.

The production of lipases by *S. aureus* was first documented more than a century ago[46]. Ever since, evidence of the requirement for bacterial lipases during *S. aureus* infection has been accumulating. For instance, anti-lipase IgG antibodies have been detected in patients infected with *S. aureus*[47]. Furthermore, the expression of lipase-encoding genes has been demonstrated during *S. aureus* infection in a murine renal abscess model[48]. However, only a handful of studies could show diminished virulence for lipase-deficient mutants in mice infected with *S. aureus*[13,38,49]. Moreover, numerous studies

have used strains with prophage-disrupted-*lip2* to successfully establish murine models of infection with *S. aureus*[8,50]. In sum, while it is reasonable to perceive lipases as virulence factors, rigorous testing in various models is still needed to fully understand the role played by *S. aureus* lipases during colonization/infection. Our data suggests that suitable environmental lipids are needed to illuminate the versatility of *S. aureus* lipases.

Our animal model relies on inflammation provoked by extensive tape-striping to enable increased colonization by *S. aureus*[51]. However, tape-stripping deplete lipids from the skin[40,41], which may explain, at least partly, the heightened colonization by *S. aureus* in this model[51]. Indeed, mice defective in sebum production and AFA synthesis take longer than wild-type mice to clear skin infection with *S. aureus*[52]. In our colonization model, we previously could not see any difference in between wild-type USA300 JE2 and Δlip in their capacity to colonize mouse skin[21]. We surmised that depleted lipids caused this absence of phenotype. Therefore, in this study, we reconstituted the lipid barrier of the mouse skin with sapienic acid (SA), or cholesterol plus SA during colonization with either WT or Δlip. We acknowledge that our model is artificial and does not recapitulate the lipid landscape of the

**Fig. 6 | Cholesterol does not prevent membrane-damaging effects of AFAs. a** Wild-type USA300 JE2 and its isogenic Δlip mutant with pEmpty, and Δlip complemented with p*lip1*, p*lip2*^S412A^, or p*lip2* were left untreated or treated with dehydroergosterol (DHE). After washing with PBS, DHE-binding was quantified by fluorometry in relative fluorescence units (RFU). **b** WT pEmpty, Δlip pEmpty, and Δlip p*lip2* were stained with azide fluor 488 upon 20-min incubation in plain nutrient broth (NB), or NB supplemented with palmitoleic acid (PA) alkyne or PA alkyne + cholesterol (Chol). Mean fluorescence intensities (MFI) were determined using flow cytometry. The indicated strain (WT pEmpty, Δlip pEmpty, Δlip p*lip1*, or Δlip p*lip2*) was incubated for 30 min (**c**) or 24 h (**d**) in NB, or NB supplemented with 50 μM PA or 50 μM PA + 50 μM Chol prior to staining with DiOC_2(3) (3,3′-diethyloxacarbocyanine iodide). Membrane potential, as computed by the ratio between red and green fluorescence intensities (red shift), was determined by fluorometry. Shown are means plus SEM for three (**a**, **b**, **d**) or four (**c**) biological replicates. Two-way ANOVA with Tukey's multiple comparisons test was used to calculate statistical significance (**b**). *$p < 0.05$, **$p < 0.01$, ***$p < 0.001$, ****$p < 0.0001$.

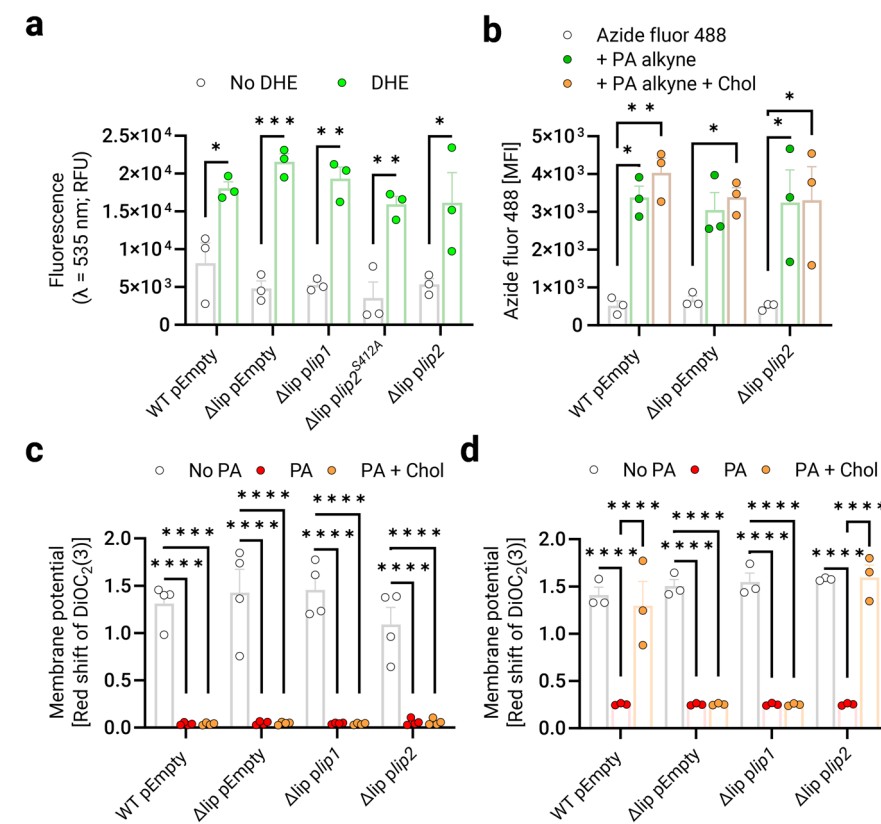

**Fig. 7 | Cholesterol contributes to the protection of *S. aureus* from AFAs on the skin. a** The occurrence of *lip2*, as detected via in silico PCR, is displayed according to the isolations site of the *S. aureus* genomes in our database. **b** The sequences of *lip2* (retrieved in **a**) were analyzed for prophage insertion. **c**, **d** USA300 JE2 (WT) and its isogenic Δlip mutant were used to topically colonize the skin of mice co-treated with cholesterol (Chol) and/or sapienic acid (SA). Five mice per group correspond to 9 or 10 skin punches, which were strongly vortexed to dislodge surface-attached bacteria (**c**), and then minced to release bacteria located in the deeper skin tissue (**d**). Viable bacteria were counted as colony forming units (CFU). Bar graphs (**c**, **d**) are medians. Statistical significance was evaluated by Kruskal–Wallis test with Dunn's multiple comparisons. **$p = 0.0013$.

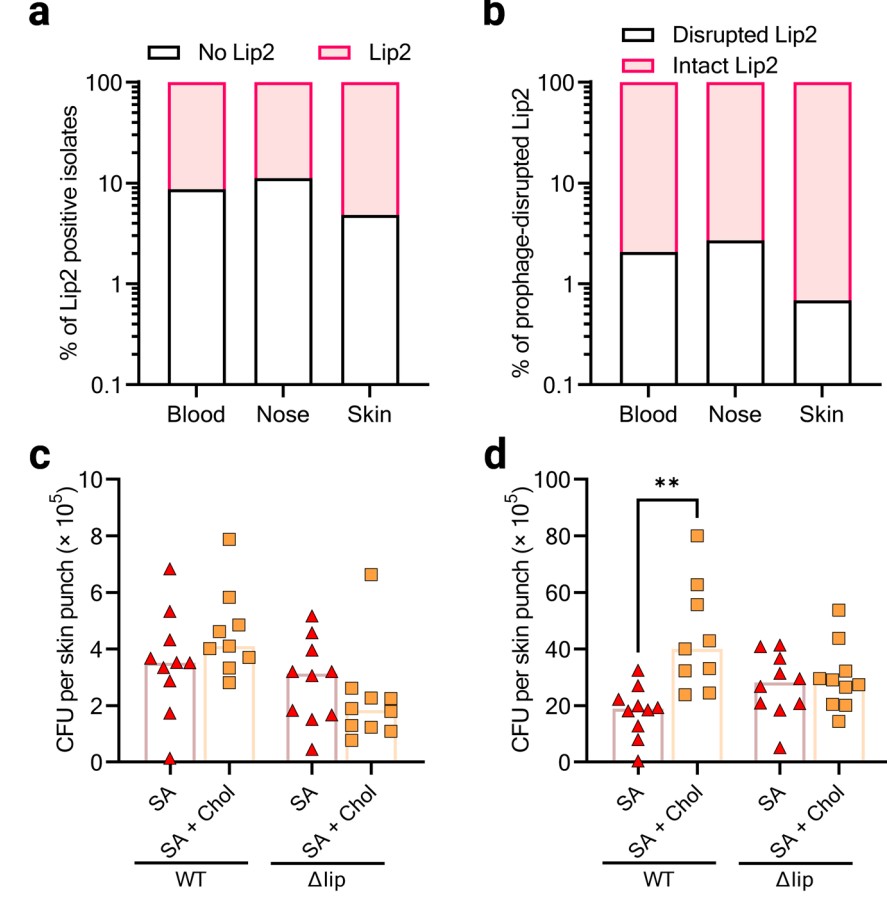

mouse skin, which includes triacylglycerols[52]. However, to the best of our knowledge, we provide the very first evidence that topical application of SA impacts skin colonization by *S. aureus*. Although we do not demonstrate that fatty acids are esterified by Lip2 on the skin, our data suggest cholesterol-mediated protection of bacteria from SA toxicity. Skin infection with *S. aureus* has been previously shown to induce the production of cholesteryl esters[52].

Cholesteryl esters appear to be especially well hydrolyzed by Lip2 (Geh) in PBS (pH 7.4) or fresh TSB (pH 7.3)[1,53]. Contrastingly, Kumar and co-workers showed that media conditioned by *S. aureus* strains with high lipolytic activity led to profound changes in bovine heart lipids, including the production of cholesteryl esters[54]. These media were collected after 6 h of bacterial growth in TSB, and probably had an acidic pH, as described elsewhere[55]. The optimum FAME (esterification) activity has been previously described to occur between pH 5.5 and pH 6[18], a pH range that is associated with heightened colonization rates by *S. aureus* in atopic dermatitis patients[56]. Little FAME activity has been observed at pH 8[18]. Interestingly, pH 8 is optimal for the lipase (hydrolysis) activity of Lip2 (Geh), which is negligible at acidic pH[23]. We show that Lip2-mediated esterification of cholesterol with fatty acids at pH 6 is residual at pH 8. It is unclear how pH dictates the reversible activity of Lip2. However, since free fatty acids are known to acidify environmental pH[57], a decrease in pH may mimic conditions with high fatty acid concentrations, which would inhibit further release of fatty acids and favor esterification activity of Lip2. Such a product-mediated regulation of enzyme activity is not unprecedented for bacterial lipases[58].

In *S. aureus*, the expression of lipase-encoding genes is controlled by the global regulators Agr and SarA[27,59]. Accordingly, the secretion of lipases Lip1 and Lip2 is impaired in mutants defective in Agr and/or SarA[60]. In a similar manner, FAME production is drastically impaired in mutants deficient in Agr or SarA[61]. In addition to a rather similar regulation, a strong correlation between lipase and FAME activities, i.e., esterification of fatty acids, has been observed for *S. aureus*[18] and some coagulase-negative staphylococci[29]. Our study provides evidence that Lip2 is the lipase catalyzing the esterification of AFAs with cholesterol. Lip2 can also use ethanol for AFA esterification whereas Lip1-mediated esterification of AFAs only took place with ethanol. This mirrors the substrate preference of Lip1 and Lip2 for short-chain and long-chain fatty acids, respectively[23,46]. It is yet unclear which structural features dictate substrate preference and activity in *S. aureus* lipases. We surmise that these features also govern the utilization of cholesterol by Lip2, which could represent a novel therapeutic target.

Esterification activity has been described for other lipases of bacterial pathogens. For instance, the lipase of *Aeromonas hydrophila* can carry out acyl transfer from glycerophospholipids to acyl acceptors, including cholesterol[55]. This glycerophospholipid-cholesterol acyltransferase activity has been demonstrated for Ssej, a virulence factor of *Salmonella* Typhimurium[62,63]. Esterification activities have also been documented for the VPA0226 lipase of *Vibrio parahaemolyticus*[64] and the ValDLT lipase of *V. alginolyticus*[65]. VPA0226 preferentially utilizes polyunsaturated fatty acids to esterify cholesterol and thereby weakens the host cell membrane to allow the escape of intracellular bacteria[64]. Similarly, lipases from *V. cholerae*[66] and *V. vulnificus*[67] have also been shown to target the membrane of host cells, although it is unclear which activity (lipase or transferase) results in cell lysis. Structural insights into ValDLT complexes with fatty acids suggest that the substrate bound to the lipase/transferase (e.g.,: saturated vs. unsaturated fatty acid) may tip the activity of the enzyme toward hydrolysis or esterification[65].

Lip2 (Geh) exhibits potent immunomodulatory properties by deacylating *S. aureus* lipoproteins to dampen the TLR2-mediated host innate immune responses[13]. However, Lip2 has been recently shown to fail at preventing macrophage activation by cell-free supernatants of *S. aureus*, when bacteria were grown in media supplemented with unsaturated fatty acids[68]. Bacterial lipoproteins purified from such cultures comprise unsaturated acyl chains that are resistant to Lip2-mediated ester hydrolysis[68]. Thus, lipolysis may not be the default function of Lip2 in an environment rich in unsaturated AFAs. Collectively, with our newfound understanding of Lip2 activities, it is enticing to posit that staphylococcal lipases play an underappreciated role in shaping host-derived lipids on the skin and at

mucosal surfaces. Eavesdropping on the lipid-mediated crosstalk between microbiomes and hosts could prove pivotal for a better understanding and prevention of colonization by opportunistic pathogens.

## Methods

### Bacterial strains and growth conditions

Bacterial strains and plasmids used in this study are detailed in Table 1. *S. aureus* and *Escherichia coli* strains were routinely grown overnight at 37 °C in tryptic soy broth (TSB) or lysogeny broth (LB), respectively. Whenever appropriate, the medium was supplemented with ampicillin (100 µg/ml), kanamycin (30 µg/ml), or chloramphenicol (10 µg/ml).

### Construction of strains

Primers used are listed in Table 2. In-frame deletion of *lip1* or *lip2* was performed with pIMAY as described previously[69]. Gene deletion was confirmed by PCR and sequencing. For mutant complementation experiments, empty pALC2073[70] (pEmpty), pALC2073-*lip1* (p*lip1*), and pALC2073-*lip2*[23] (p*lip2*) were used. To generate p*lip2*$^{S412A}$, p*lip2* was amplified with mutagenic primers. *E. coli* IM08 was then transformed with the DpnI-treated PCR product. After plasmid purification, successful mutagenesis was confirmed by digestion with PaeI and sequencing.

### Purification of recombinant lipases

N-terminally His$_6$-tagged Lip2 or Lip2 S412A without signal peptide[23] was overexpressed in *E. coli* BL21 (DE3). After cell lysis, immobilized metal affinity chromatography (IMAC, Biorad) with nickel resin was utilized to purify recombinant protein according to standard procedures[13].

### Membrane vesicle purification

MVs were isolated with the ExoQuickTC reagent (EXOTC10A-1; System Biosciences) as described elsewhere[20,71]. Briefly, bacteria grown overnight were diluted to an optical density at 600 nm of 0.1 (OD$_{600}$) in 20 ml fresh TSB and grown with shaking for 6 h (late exponential growth phase). After centrifugation, supernatants were sterile filtered and concentrated with 100-kDa centrifugal concentrator cartridges (Vivaspin 20; Sartorius) prior to precipitation with ExoQuickTC and resuspension in phosphate-buffered saline.

### Growth assays

Growth assays were performed in TSB (Oxoid), nutrient broth no.2 (NB; Oxoid), basic medium (BM: 1% soy peptone, 0.5% yeast extract, 0.5% NaCl, 0.1% glucose and 0.1% K$_2$HPO$_4$, pH 7.2), or chemically defined medium[72] as described previously[20]. Overnight bacterial cultures (OD$_{600}$ of ~13) or bacteria grown to the exponential phase OD$_{600}$ of ~2.5) were diluted to an OD$_{600}$ of ~0.01 in plain medium or medium supplemented with AFAs (50 to 200 µM), cholesterol (50 to 100 µM), MVs (1 µg/ml), and/or recombinant Lip2 (1 µg/ml). Bacteria were then grown in a 96-well plate (U-bottom) at 37 °C with linear shaking at 567 cpm (3-mm excursion) for 24 h. The OD$_{600}$ was measured every 15 min with an Epoch 2 plate reader (BioTek). Areas under growth curves were computed with GraphPad Prism 9.5.1.

### Biofilm assay

Biofilms formed under static conditions at 37 °C for 24 h in cell culture 24-well plates (Greiner) were stained with safranin as described elsewhere[73]. Unbound safranin was washed with PBS, and biofilm-associated safranin was incubated with 70% ethanol and 10% isopropanol for elution. A CLARIOStar microplate reader (BMG Labtech) was used to measure OD$_{530}$ and quantify biofilms.

### Lipase activity assay

The lipase activity of bacteria-conditioned media was assayed with *para*-nitrophenyl palmitate as previously described[23]. Bacteria-conditioned media were diluted fifty times with the assay buffer (50 mM Tris·HCl, 0.005% Triton X-100, 1 mg/ml gum arabic at pH 8.0) supplemented with 0.8 mM *para*-nitrophenyl palmitate. After incubation at 37 °C for 30 min, OD$_{405}$ was measured with a CLARIOStar microplate reader (BMG Labtech).

## Table 1 | Strains and plasmids

| Strain or plasmid | Description | Source or reference |
|---|---|---|
| *Staphylococcus aureus* | | |
| Newman | Methicillin-sensitive *S. aureus* (MSSA) | 83 |
| SH1000 | NCTC8325 derivative with a functional *rsbU* gene, Δ*tcaR*, cured of φ11, φ12, and φ13; MSSA | Simon Foster[84] |
| USA300 | Community-acquired MRSA (CA-MRSA), plasmid-cured derivative of LAC strain | David E. Heinrichs[23] |
| USA300 Δ*lip2* | USA300 with the *lip2* (*gehB*) gene deleted | David E. Heinrichs[23] |
| USA300 JE2 | CA-MRSA, plasmid-cured derivative of LAC strain | Paul Fey[85] |
| USA300 JE2 Δlip | USA300 JE2 defective for Lip1 and Lip2 | Friedrich Götz[21] |
| USA400 MW2 | CA-MRSA | Michael Otto[86] |
| USA400 MW2 Δ*lip1* | MW2 with the *lip1* (*gehA*) gene deleted | This study |
| USA400 MW2 Δ*lip2* | MW2 with the *lip2* (*gehB*) gene deleted | This study |
| *Escherichia coli* | | |
| BL21 (DE3) | F⁻ *ompT hsdS*$_B$(r$_B^-$ m$_B^-$) *gal dcm* (DE3), IPTG-inducible T7 RNA polymerase | New England Biolabs |
| DC10B | Δ*dcm* in the DH10B background (K-12 derivative) | Simon Heilbronner[69] |
| SA08B | DC10BΩP$_{help}$-*hsdMS* (CC8-2) of NRS384 integrated between the *atpI* and *gidB* genes | Simon Heilbronner[87] |
| IM01B | SA08BΩP$_{N25}$-*hsdS* (CC1-1) of MW2 integrated between the *essQ* and *cspB* genes | Simon Heilbronner[87] |
| Plasmid | | |
| pIMAY | Temperature-sensitive allelic exchange plasmid for staphylococci | Simon Heilbronner[69] |
| pIMAY-Δ*lip1* | pIMAY carrying the *lip1*(*gehA*) deletion construct. | This study |
| pIMAY-Δ*lip2* | pIMAY carrying the *lip2* (*gehB*) deletion construct. | This study |
| pEmpty (pALC2073) | *E. coli-S. aureus* shuttle vector; contains Pxyl/tet | David E. Heinrichs[70] |
| p*lip1* | pALC2073 containing the *lip1* coding region | This study |
| p*lip2* | pALC2073 containing the *lip2* coding region | David E. Heinrichs[23] |
| p*lip2*$^{S412A}$ | pALC2073 containing the mutated *lip2* coding region to generate Lip2 S412A | This study |
| pET28a(+)-*lip2* | pET28a(+) for Lip2 expression | David E. Heinrichs[23] |
| pET28a(+)-*lip2*$^{S412A}$ | pET28a(+) for Lip2 S412A expression | David E. Heinrichs[23] |

## Table 2 | Primers

| Oligonucleotide | Sequence (5'—3') |
|---|---|
| For deletion of *lip1* (pIMAY-Δ*lip1*): | |
| KpnI-*lip1* ups-fwd | CTAGAAGGTACCGGTGTCGGCATGATATTGCG |
| *lip1* ups-rev | CAAGCATAATTTATAAAGTAAAGGGAGG |
| SOEING-*lip1* dwn-fwd | CCTCCCTTTACTTTATAAATTATGCTTGACTTTTCATCATTGTCAGCACCTC |
| SacI-*lip1* dwns-rev | GCATCCGAGCTCGTAGGATACTTACTTTGAGGGAAG |
| For deletion of *lip2* (pIMAY-Δ*lip2*) | |
| KpnI-*lip2* ups-fwd | GTAGAGGTACCGTATGCCCACTAAACTATAGAC |
| *lip2* ups-rev | TCCTCTTAACATATAATCACCTC |
| SOEING-*lip2* dwn-fwd | GAGGTGATTATATGTTAAGAGGAGCAAGTTAAATTCATCTTCTG |
| SacI-*lip2* dwns-rev | CATGTTGAGCTCCGTTTATGCACGTGGCACAGG |
| For site-directed mutagenesis of p*lip2* into p*lip2*$^{S412A}$ | |
| *lip2*$^{S412A}$-fwd | GTAGGGCATGCTATGGGTGG |
| *lip2*$^{S412A}$-rev | CAAGATGTACCTTTTTACCAGG |

### Cholesterol quantification with Amplex™ red cholesterol assay kit

Equimolar concentrations (300 μM) of cholesterol and linoleic acid were added to 0.1 M sodium phosphate buffer (pH 6). Free cholesterol was then quantified before (timepoint zero) or after the addition of recombinant Lip2 to samples and incubation at 37 °C with shaking. To that end, Amplex™ red cholesterol assay kit was used according to the manufacturer's instructions.

### FAME activity assay, lipid extraction and HPTLC

Recombinant lipases (1.4 ng/ml) or bacteria-conditioned media were diluted in 0.1 M sodium phosphate buffer (pH 6 unless stated otherwise) supplemented with equimolar concentrations (0.3 to 1.2 mM) of AFAs and cholesterol dissolved in DMSO or acetone, respectively. Upon overnight (18–22 h) incubation in glass vials at 37 °C with shaking, methanol (MeOH) and chloroform were added to stop the reaction and extract lipids according to the Bligh and Dyer protocol[74]. The organic fraction was transferred to a fresh vial, dried, and resuspended in 2:1 (vol/vol) chloroform: MeOH. Lipid extracts were then applied to silica gel high-performance thin-layer chromatography (HPTLC) plates (silica gel 60 F$_{254}$, Merck) using a Linomat 5 sample application unit (CAMAG). Plates were developed in an automatic developing chamber ADC 2 (CAMAG) with a mobile phase system 90:10:1 (vol/vol/vol) petroleum ether: ethyl ether: acetic acid[64]. Lipid spots were visualized in an iodine vapor chamber.

## Internal standards and chemicals used for lipid analysis by untargeted UHPLC MS/MS

EquiSPLASH™ LIPIDOMIX® quantitative mass spectrometry internal standard, phosphatidic acid 15:0-18:1 (d7), cholesterol (d7), cholesteryl ester (CE) 18:1 (d7), lyso sphingomyelin (LSM) d18:1 (d9) and palmitoyl-L-Carnitine (CAR) 16:0 (d3) were obtained from Avanti Polar Lipids (Alabaster, AL, USA). Arachidonic acid (AA) (d11) and ceramide (Cer) d18:1-15:0 (d7) were purchased from Cayman Chemicals (Ann Arbor, MI, USA). Isopropanol (IPA), acetonitrile (ACN) and methanol (MeOH) in Ultra LC-MS grade were from Carl Roth (Karlsruhe, Germany). Ammonium formate, formic acid and IPA in HPLC grade were purchased from Merck (Darmstadt, Germany). Purified water was produced by Elga Purelab Ultra (Celle, Germany).

## Sample preparation for lipid analysis by UHPLC MS/MS

Prior to lipid extraction, a mixture of internal standards was prepared by mixing ice-cold MeOH with LIPIDOMIX®, phosphatidic acid 15:0–18:1 (d7), cholesterol (d7), CE 18:1 (d7), LSM d18:1 (d9), CAR 16:0 (d3), AA (d11), and Cer d18:1-15:0 (d7). This internal standard mixture (225 μl) was then added to each sample. Lipid extraction was then performed according to a biphasic extraction method[75,76]. Samples supplemented with standards were vortexed for 10 s. Next, 750 μl ice-cold methyl tert-butyl ether (MTBE) was added to each sample. After 1h-incubation on ice, each sample was supplemented with water (185 μl) to obtain a final ratio of 10:3:2.5 (vol/vol/vol) for MTBE, MeOH, and water, respectively. Samples were then incubated at room temperature for 10 min to induce phase separation. The upper (organic) phase was transferred to a fresh tube. MTBE:MeOH:water (10:3:2.5; vol/vol/vol) was added to the lower (water) phase for re-extraction of lipids. The upper phase from the second extraction was then combined with the upper phase from the first extraction. The combined extracts were evaporated to dryness with GeneVac EZ2 evaporator (Ipswich, UK) under nitrogen protection. Lipid films were reconstituted in 100 μl MeOH. After vortexing (10 s), sonication (2 min), and centrifugation (10 min, $3500 \times g$), lipid extracts were transferred to autosampler vials.

A pooled quality control (QC) sample was prepared by mixing 15 μl of each re-constituted sample.

## Lipid analysis by UHPLC MS/MS

Samples were analyzed with an Agilent 1290 Infinity UHPLC system (Agilent, Waldbronn, Germany) equipped with a binary pump, a PAL-HTX xt DLW autosampler (CTC Analytics AG, Switzerland) and coupled to a SCIEX TripleTOF 5600 + quadruple time of flight (QTOF) mass spectrometer with a DuoSpray Source (SCIEX, Ontario, Canada). The chromatographic separation was performed on an ACQUITY UPLC CSH C18 column (100 mm × 2.1 mm; particles: 1.7 μm; Waters Corporation, Millford, MA, USA) with precolumn (5 mm × 2.1 mm; 1.7 μm particles). The column temperature was 65 °C with a flow rate of 0.6 ml/min. Mobile phase A was composed of water: acetonitrile (2:3; vol/vol) supplemented with 10 mM ammonium formate and 0.1% formic acid (vol/vol). The mobile phase B was IPA:ACN:water 90:9:1 (vol/vol/vol) containing 10 mM ammonium formate and 0.1% formic acid (vol/vol). A gradient elution started from 15% B to 30% B in 2 min, followed by increase of B to 48% in 0.5 min. Mobile phase B was then further increased to 82% at 11 min and quickly reached 99% in the next 0.5 min, followed by holding this percentage for another 0.5 min. Afterwards, the percentage of B was switched back to starting conditions (15% B) in 0.1 min to re-equilibrate the column for the next injection (2.9 min).

UHPLC-MS/MS experiments were operated in both positive and negative mode with injection volumes of 3 μl for positive and 5 μl for negative mode. An MS full scan experiment with mass range *m/z* 50 to 1250 was selected, while different SWATH windows were acquired for MS/MS experiments (Supplementary Table S2). The ion source temperature was set to 350 °C with curtain gas, nebulizer gas and heater gas pressures 35 lb/in², 60 lb/in², and 60 lb/in², respectively, for both modes. The ion spray voltage was set to 5500 V in the positive mode and −4500 V in negative mode. The declustering potential was adjusted to 80 V and −80 V for positive and negative polarity mode, respectively. The cycle time was always 720 ms. The collision energy and collision energy spread for each experiment are specified in detail Supplementary Table S2.

The sequence was started with three injections of internal standard mixture as system suitability test followed by blank extract and QC sample. The whole sequence was controlled by injection of QC sample after every five samples to monitor the performance of the instrument throughout the analytical batch.

## Binding assays with DHE

Upon centrifugation, bacteria grown overnight ($OD_{600}$ of 0.7 per sample) were resuspended in 1 ml NB and incubated for 30 min at 37 °C with shaking. Samples were left untreated or treated with 20 μM dehydroergosterol (DHE, Sigma) for an additional 30 min at 37 °C with shaking. Next, bacteria were washed with PBS before their fluorescence (excitation wavelength = 324 nm, emission wavelength = 535 nm) was measured and normalized to their $OD_{600}$.

## Click chemistry with palmitoleic acid alkyne

Click chemistry with an AFA analog was performed as described elsewhere[20]. Overnight bacterial cultures were centrifuged ($OD_{600}$ ~ 1.4 per sample), resuspended in 200 μl NB or NB + 50 μM cholesterol, and incubated at 37 °C for 20 min with 40 μM palmitoleic acid alkyne (Cayman Chemical). After centrifugation, bacterial pellets were resuspended in Click-iT cell reaction buffer supplemented with copper(II) sulfate and Click-iT cell buffer additive, as per manufacturer's recommendations (Click-iT cell reaction buffer kit; Invitrogen). Click chemistry was performed at 25 °C for 30 min with 7 μM azide fluor 488 (Merck). After washing with PBS, bacteria were analyzed by flow cytometry (BD LSRFortessa flow cytometer).

## Membrane potential measurements

Overnight bacterial cultures were centrifuged ($OD_{600}$ of 0.7 per sample). Bacteria were then resuspended in 1 ml plain NB or NB supplemented with 50 μM PA alone or with 50 μM cholesterol. The samples were incubated for 30 min at 37 °C with shaking. After the addition of 10 μl of the membrane potential probe, 3,3′-diethyloxacarbocyanine iodide [$DiOC_2(3)$] to each sample, bacteria were incubated for 30 min at 37 °C with shaking. Next, bacteria were centrifuged and resuspended in PBS. The green fluorescence (excitation wavelength = 470 nm, emission wavelength = 515 nm) and the red fluorescence (excitation wavelength = 550 nm, emission wavelength = 605 nm) were measured with a CLARIOStar microplate reader (BMG Labtech). The membrane potential was computed as the ratio red/green fluorescence. Alternatively, bacteria were grown for 24 h in a 96-well plate (U-bottom) at 37 °C before staining with [$DiOC_2(3)$] and membrane potential measurements.

## Genomic analyses

For our custom database, *S. aureus* genomes (3835) downloaded from the BV-BRC[36]. After manually curating the metadata, the database was stratified to blood (1481), nose (1587) and skin (767) according to isolation sites. To extract *lip2* gene sequence, in silico PCR was performed with a Perl script (https://github.com/egonozer/in_silico_pcr) using forward and reverse primers 5'-ATGTTAAGAGGACAAGAAGAAA-3' and 5'-TTAACTTG CTTTTCAATTGTGTT-3', respectively, and allowing 5 mismatch/indels. To detect prophages, results of the in silico PCR were uploaded as one multi-FASTA file to PHASTER (https://phaster.ca/)[77]. Prophage-disrupted *lip2* amplicons were not included during the subsequent analysis pertaining to amino acid sequence variability of Lip2. Extracted sequences from the in silico PCR were translated into full-length Lip2 protein sequences and aligned using MAFFT (v7.310)[78] with default parameters using Lip2 sequence from *S. aureus* USA300 strain FPR3757 (accession number NC_007793.1) as reference. A Python script (https://github.com/AhmedElsherbini/Align2XL) was then used to extract mutation rates from the aligned protein sequences. Alignmentviewer was utilized to represent Lip2 multiple sequence alignment in two-dimensional space using

the Uniform Manifold Approximation and Projection (UMAP) dimensionality reduction algorithm[79].

## Mouse experiments

C57BL/6 mice (Envigo) were colonized epicutaneously with *S. aureus* following tape-stripping as described previously[21,39]. Briefly, overnight cultures of USA300 JE2 or its isogenic Δlip mutant were washed twice with PBS and adjusted to $5 \times 10^9$ cells per ml. An inoculum of 15 µl from the washed bacterial suspension was added to a film paper disc. In addition to bacteria, these discs were supplemented with cholesterol (7 µg) and/or sapienic acid (5 µg). Two discs with bacteria and lipids per mouse were placed onto the back skin that had been shaved and tape-stripped seven times to facilitate *S. aureus* establishment. Finn chambers on Scanpor (Smart Practise, Phoenix, AZ, USA) and plasters (Tegaderm) were used to fix discs on mouse back skin. After 24 h with frequent monitoring, Finn chambers were removed, mice were euthanized, and a biopsy puncher was used to collect *S. aureus*-colonized skin. These skin punches were vortexed in PBS for 30 s to dislodge surface-attached bacteria. Skin punches were then minced with scalpels and homogenized by vortexing for 30 s in PBS to release tissue-associated bacteria. Surface associated and tissue-associated bacteria were enumerated following serial dilution with PBS, plating on tryptic soy agar, and incubation overnight at 37 °C.

## Statistics and reproducibility

Statistical tests, which are all specified in the figure legends, were performed with Prism 9.5.1 (GraphPad), and *p* values < 0.05 were considered significant. Analysis of variance (ANOVA) with Dunn's, Dunnett's, Šídák's, or Tukey's multiple-comparison test was used. In vitro assays involving bacteria were performed with at least three different cultures ($n = 3$ to 5 biological replicates), which were inoculated on separate days. Experiments with recombinant Lip2 were performed with proteins purified independently from two different cultures, and assays were repeated on three separate occasions ($n = 3$ experimental replicates). The in vivo assay was conducted with one bacterial culture per strain (WT USA300 JE2 or Δlip) and colonized mice were considered to be the source of biological variability ($n = 9$ or 10).

## Ethics statement

All experimental procedures involving mice were carried out according to protocols approved by the Animal Ethics Committees of the Regierungspräsidium Tübingen (IMIT 3/18G).

## Reporting summary

Further information on research design is available in the Nature Portfolio Reporting Summary linked to this article.

## Data availability

Amplicons of the in silico PCR of lip2 genes (https://doi.org/10.6084/m9.figshare.23828196.v1)[80], multiple sequence alignment of Lip2 (https://doi.org/10.6084/m9.figshare.23828274.v1)[81] and prophage sequences within *lip2* (https://doi.org/10.6084/m9.figshare.23828274.v1)[82] are available on Figshare. All the other data are available within the article and its Supplementary Materials, which include Source data (Supplementary Data 1).

## Materials availability

Bacterial strains generated in this study are available from the corresponding author.

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

## Acknowledgements
We thank David E. Heinrichs (University of Western Ontario), Paul Fey (University of Nebraska Medical Center), and Friedrich Götz (University of Tübingen) for providing us with bacterial strains. We are indebted to Dr Libera Lo Presti (University of Tübingen) for critical feedback on the manuscript, and to Ulrike Redel for technical support. We acknowledge support by the High Performance and Cloud Computing Group at the Zentrum für Datenverarbeitung (University of Tübingen), the state of Baden-Württemberg through bwHPC and the Deutsche Forschungsgemeinschaft (DFG) through the grant INST 37/935-1 FUGG. A.K.T. is recipient of a fellowship from the Alexander von Humboldt Foundation. X.F. gratefully acknowledges the support from the China Scholarship Council (grant number 201908080155). This work was supported by grants from the DFG via the Cluster of Excellence EXC 2124 "Controlling Microbes to Fight Infections" project ID 390838134 to A.K.T., B.S., and A.P. We acknowledge support from the Open Access Publication Fund of the University of Tübingen.

## Author contributions
Conceptualization: A.K.T.; Methodology: A.K.T., A.M.A.E., B.S., J.C., X.F., and D.K.; Validation: J.C., L.S., M.A.B., O.G., and S.P.L.; Investigation: A.K.T., A.M.A.E., X.F., J.C., O.G., L.S., M.A.B., X.H., M.Le., A.H., and D.K.; Resources: A.K.T., B.S., M.Lä., and A.P.; Writing—original draft: A.K.T.; Writing—review & editing: A.K.T., M.Lä., M.A.B., D.K., and A.P.; Visualization: A.K.T., A.M.A.E., J.C., and X.F.; Funding acquisition: A.K.T., B.S., M.Lä., and A.P.; Project administration: A.K.T.; Supervision: A.K.T., D.K., M.Lä., and A.P.

## Funding

## Competing interests
The authors declare no competing interests.
