## [Peer review file · Communications Biology]

Reviewers' comments:

Reviewer #1 (Remarks to the Author):

In this manuscript, the authors identified the *S. aureus* lipase Lip2 as a resistance factor against antimicrobial fatty acid (AFA) and demonstrated that Lip2 detoxifies the palmitoleic AFA toxicity via one-step esterification of cholesterol *in vitro*. The authors also utilized a mice skin colonization model to reveal that wildtype *S. aureus* survived better on the skin from sapienic acid when additional cholesterol was also applied, whereas the Lip mutant showed lower growth advantage under the same experimental conditions. Overall, the authors demonstrate the esterification enzymatic activities of Lip2 against AFA, as a self-protection mechanism against AFA from *S. aureus*.

The main limitation of the study is whether or not the esterification activities and substrate specific described herein of Lip2 are biologically relevant in light of the high concentrations used in the *in vitro* studies and the artificial nature of the *in vivo* study.

Additional comments:

1. The medium used in all the growth curve and survival studies are not chemically defined media. Although it seems that wt *S. aureus* and lip mutants have no growth differences when no PA and cholesterol are introduced, these rich media could introduce artifacts due to the rich nutrients and growth factors within the media. Have the authors considered testing the growth conditions with PA and cholesterol in chemically defined medium and see if Lip2 is truly able to rescue *S. aureus* from AFA via esterification with cholesterol?
2. The dose of both cholesterol and linoleic acid used in the *in vitro* enzymatic experiment in Fig. 5 are relatively high (300 μ M). Under this single-dose condition, it is difficult to determine whether the esterification from Lip2 is truly due to the substrate-specific catalytic activity of purified Lip2 enzyme, or this esterification was catalyzed by Lip2 because both the LA and cholesterol concentration was set to a high. The authors should consider doing additional titration experiments to demonstrate the catalytic activities from Lip2 is truly dose-dependent and specific to both substrates. Also, have the authors tested the possible esterification activities of purified Lip1?
3. On Fig. 5A., as the time proceeded to 120 min, the cholesterol concentration stopped decreasing around 20-25 μ M. Is this because of the loss of Lip2's enzymatic activity after 2 hours? Or a minimum concentration of 20-25 μ M cholesterol is essential for the Lip2 to esterify the AFA?
4. The *in vivo* data is modest and artificial. Why mice needed supplementation of cholesterol and sapienic acid? Do mice produce antimicrobial fatty acid?
5. From Fig. 1 to Fig. 4, the authors tested the growth advantage of wt *S. aureus* growing in medium supplemented with PA or PA + cholesterol combination. Besides the 50 μ M inhibition concentration of PA mentioned in line 97 of the manuscript, the rest of the experiment in Fig. 1 to Fig. 4 do not clearly listed the actual concentration of PA and cholesterol used in the main text and figure caption. (However, the concentration of PA and cholesterol was both clearly listed on the SI section.) The

main text should also have all the concentration clearly listed. Also, it is important to provide additional rationales of how the PA, LA and cholesterol concentration used are biologically relevant.

Reviewer #2 (Remarks to the Author):

In this manuscript, Tchoupa et al study the role of two *Staphylococcus aureus* secreted lipases in the resistance to antimicrobial fatty acids (AFAs). The authors conduct compelling and well-controlled in vitro studies to demonstrate that AFAs are detoxified when *S. aureus* is grown in the presence of cholesterol. AFA detoxification in the presence of cholesterol was dependent on the Lipase Lip2 (also known as Geh or Sal2), but not Lip1 (Sal1). Lip2 is a well-characterized glycerol ester hydrolase that can hydrolyze ester linked fatty acids from triglycerides, but also lipoprotein, phospholipids, and cholesterol esters. Several lipases of this class are known to also exhibit reversible esterification activity, whereby they form fatty acid esters with cholesterol or other alcohols (FAME activity). *S. aureus* is known to produce an enzyme with FAME activity, but the enzyme responsible has remained elusive. Because Lip2 was able to detoxify antimicrobial fatty acids in the presence of cholesterol in a Lip2 dependent manner, the authors reasoned this might be due to FAME activity of Lip2 leading to the formation of less toxic cholesterol esters. The authors use in vitro biochemical assays along with TLC and Mass spectrometry to show that this is true. They then follow-up with a series of studies to ascertain how the presence of cholesterol and lip2 might be preventing toxic effects on bacterial cells and find that cholesterol and lipid binding to the cell surface is unchanged and that membrane potential is lost regardless of lipase presence. Lastly, the authors survey several thousand *S. aureus* isolates and find that Lip2 is well conserved with relatively low levels of prophage insertional disruption. Although low, there was a difference in the number of strains with prophage insertion from the blood/nose (~2%) versus the skin (~1%), which might indicate a role in skin colonization. The authors try to conduct an experiment to get at a role for Lip2 in promoting skin colonization and find that addition of sapienic acid and cholesterol along with *S. aureus* to the skin seems to promote survival of lip2 containing strains within the skin tissue.

Overall, this study is well-controlled and clearly demonstrates that Lip2 has FAME activity and promotes survival in the presence of antimicrobial fatty acids when grown with cholesterol in vitro (Figures 1-5). These figures represent a major strength of the study. However, the conclusions related to figures 6 and 7 are not felt to be fully justified at this time. In addition, the discussion does not fully place this work within the context of the literature on known lipases with FAME activity or provide a compelling synthesis on the potential for the reversible activity of Lip2 (e.g. when might hydrolysis predominant over esterification etc and would would the advantage be based on recent studies in the literature).

1. Figure 6 - Using cholesterol and fatty acid binding studies along with disruption of membrane potential the authors reason that cholesterol ester formation is the only plausible means of protection from AFA in the presence cholesterol. However, it is not clear how this is happening,

especially if membrane damage (loss of membrane potential) still occurs in all conditions? How exactly are AFA cholesterol esters protective? This could be more clearly described or considered in the discussion.

2. Although the difference in the percentage of strains (2.1% in blood versus 0.7% on skin) containing a phage insertion in lip2 is noted by the authors it is not clear how well this data justifies the possibility that lip2 is required for skin colonization over an equally important requirement at other sites. 99% of skin isolates do not have a disruption in lip2 versus 98% of blood and nose. It is not clear that there is a notable correlation with phage insertion that is supported by this data, except that the majority of *S. aureus* isolates do not have phage disruption regardless of site. This data might argue that the lipase is important for host microbe interactions overall since most strains lack a phage disruption, but it is not clear it says much about skin over other sites. Can the authors either justify this claim better or modify their conclusions to better reflect the results.

3. Figure 7 infection data do not really test the hypothesis the lipase mediated esterification is beneficial for *S. aureus* colonization of the skin. The experiment supplies sapienic acid and cholesterol together with *S. aureus* rather than testing the role of the lipid environment of the skin itself. As it stands there is no strong evidence that Lip2 promotes skin colonization via esterification of cholesterol to alleviate AFA stress. Conclusions from this experiment should be calibrated to reflect the strengths and weaknesses of the experimental design. This should be addressed in the discussion.

4. Reference 1 of the manuscript has previously shown that cholesterol esters themselves are substrates for ester hydrolysis by Lip2 in vitro. The ability to hydrolyze FA esters versus esterification of free FA is heavily influenced by conditions, lipid/FA, and alcohol abundance, and other conditions, especially in vitro. The authors might consider taking some time to reconcile their results with the hydrolysis activity shown in Ref 1 and provide their take on the potential for the reversible activity of Lip2 (e.g. when might hydrolysis be predominant over esterification). For example, there is an interesting paper from the Orth lab (which is already referenced in the work for methodological reasons, see 56) that highlights some of the potential diverse functions for lipase FAME activity in pathogenesis. There seems to be a lot of opportunity to put the findings of this work in a broader context and reflect on the importance of the potential of this shift from hydrolysis to esterification.

Reviewer #3 (Remarks to the Author):

This manuscript concludes that the Lip2 lipase of *S. aureus* has a reverse role in esterifying fatty acids to cholesterol, and that this function has a critical role in detoxifying antimicrobial fatty acids encountered on skin and nasal secretions. The authors provide a convincing demonstration that Lip2 accounts for the fatty acid modifying enzyme FAME activity that has been known of for many years but never identified. The biochemical data and growth analyses are supportive of a role for Lip2 in detoxifying antimicrobial fatty acids through esterification to cholesterol. The biochemical data are

strong and leaning toward excessive, but conclusions of relevance to skin colonization need to be tempered.

Major comments:

1. Data in support of lipase esterification of free fatty acid to cholesterol is convincing, as are the in vitro data indicating that toxicity of palmitoleic acid is ameliorated in the presence of cholesterol. However, on human skin, sebaceous secretions contain cholesterol, free fatty acids and triglyceride, of which the latter is most likely a preferred substrate for lipase. A question that should be addressed is whether lipase will promote esterification of free fatty acids to cholesterol in the presence of triglyceride, which is also abundant in sebaceous secretions.

2. Skin is known to have acidic pH. Have the authors assessed the influence of pH on the esterification reaction? Have previous studies on FAME activity addressed the optimum pH for this reaction?

3. Lines 260-279 describing how cholesterol does not prevent membrane-damaging effects of antimicrobial fatty acids is not clearly written. First the authors propose a potential but arguably also imaginative alternative explanation for how cholesterol might alleviate toxicity of fatty acids; namely that a secreted lipase might promote microbial binding to cholesterol such that the resulting aggregates are less susceptible to fatty acid. The authors conduct an experiment that negates this hypothesis, and then use a click chemistry experiment to suggest that cholesterol does not prevent membrane targeting of AFA's. This conclusion seems counterintuitive since cholesterol alleviates toxicity and toxicity is attributed to membrane damage. Note that the click chemistry experiment SHOULD demonstrate a significant difference in the distribution of free fatty acid, which can either be incorporated into phospholipid, esterified to cholesterol, or accumulate in the membrane as a free fatty acid, which is what promotes toxicity. As such, this entire section fails to make a strong point and detracts from the linear narrative of the story.

4. Lines 336 to 349 comprise a section entitled "The capacity to manipulate cholesterol governs skin colonization by *S. aureus*". Contrary to the certainty implied by the title of this section, the text of the results section is less emphatic, indicating that the lipase deficient mutant "was largely" unaffected by cholesterol application, while the Lip-2 proficient wild type "appeared to benefit" from cholesterol application, and the data "strongly suggest" that *S. aureus* utilizes lipase to manipulate environmental lipids. The authors are correct to imply some uncertainty as to the rigor of this conclusion.

5. Regarding the colonization model, it is recognized that it is challenging to select an appropriate model. The model that is chosen can be useful but is subject to limitations; namely that it mimics atopic dermatitis where there is a deficiency in the skin barrier, and indeed the authors acknowledge this by stating that the taping technique strips the skin of its lipids, necessitating that the lipid barrier needs to be reconstituted by addition of cholesterol and free fatty acids in order to observe a phenotype between wild type and Lip2 deficient mutant. There are a lot of caveats here. One thought to consider here is that the same taping technique that has previously been used to recover and define the lipid composition of sebaceous secretions could also be applied post-challenge to determine if there is evidence of sapienic acid esterification to cholesterol on the skin of mice. If the authors apply a cholesterol of known composition that differs from the predominant cholesterol in

the skin of mice, it should be feasible to determine if it is being modified by esterification of sapienic acid, which is not produced by mice.

6. The authors should acknowledge that Lip2 is equivalent to the glycerol ester hydrolase Geh, which was also shown to modulate the immune response by deacylating *S. aureus* lipoproteins, such that the host innate immune response mediated by TLR2 is dampened. (PNAS 2019; 116:3764-3773). While it is difficult to relate the authors current data on “colonization” to previous work on lip2 deficiency and persistence in organs, it should at least be acknowledged that the lip2/geh deficiency could alter the cytokine response to infection. Notably, the authors did not observe any impact of lip2 and cholesterol on what is presumed to be surface adherent bacteria, whereas the minced tissue samples presumably representing tissue invasion indicated significantly more bacteria in the lip2 deficient strain. Some H&E staining of the infected tissue and pathology analysis would be informative here.

Minor comments:

1. Even in the presence of cholesterol, wild type bacteria still exhibited an ~ 7.5h lag phase when cultured with palmitoleic acid compared to media alone. Consequently, it appears that palmitoleic acid does retain some toxicity in presence of cholesterol.
2. The authors use overnight (and presumably) stationary phase cultures as inoculum for the growth assays. There is no mention here of washing the cells prior to inoculating plates for growth assays. Does the 7.5h lag relate to the ability of lipase carried over in the inoculum to esterify fatty acid to cholesterol? Would the lag phase be longer if the bacteria were washed prior to inoculation? Note that for the infection experiments, the methodology specifies that the bacteria were washed prior to challenge.
3. Exponential phase cells are more susceptible to the bactericidal activity of antimicrobial fatty acids compared to stationary phase cells. How would the data look if exponential phase cells were used as inoculum?
4. As the lead figure in the paper, the choice of symbols and colors for Fig 1 is very poor, making it difficult to follow.
5. From Fig's 1A and 1D it is apparent that WT *S. aureus* exhibits a significantly lower growth rate in media supplemented with palmitoleic acid and cholesterol compared to medium alone, and also reaches a significantly lower final OD600. Consequently, it appears that there is still significant toxicity associated with the palmitoleic acid. Also, expressing data as “Area Under the Curve” does not show to what extent there is an extended lag phase, which is also a sign of stress.

Reviewers' comments (in black) and Authors' response (in blue):

Reviewer #1 (Remarks to the Author):

In this manuscript, the authors identified the *S. aureus* lipase Lip2 as a resistance factor against antimicrobial fatty acid (AFA) and demonstrated that Lip2 detoxifies the palmitoleic AFA toxicity via one-step esterification of cholesterol *in vitro*. The authors also utilized a mice skin colonization model to reveal that wildtype *S. aureus* survived better on the skin from sapienic acid when additional cholesterol was also applied, whereas the Lip mutant showed lower growth advantage under the same experimental conditions. Overall, the authors demonstrate the esterification enzymatic activities of Lip2 against AFA, as a self-protection mechanism against AFA from *S. aureus*.

We thank this reviewer for the overall positive appraisal of our work.

The main limitation of the study is whether or not the esterification activities and substrate specific described herein of Lip2 are biologically relevant in light of the high concentrations used in the *in vitro* studies and the artificial nature of the *in vivo* study.

The concentrations that we used *in vitro* to inhibit *S. aureus* in rich media (50 to 200 \$\mu\text{M}\$ ) may seem high at the first sight. However, they are similar to concentrations used in a pioneering study (40 \$\mu\text{M}\$ to 200 \$\mu\text{M}\$ ) about sapienic acid with a chemically defined metal limitation medium amongst other media (Clarke et al., 2007; doi: 10.1016/j.chom.2007.04.005). For their *in vivo* assays, Clarke and colleagues even used AFA concentrations within the molar range. This is not surprising since physiological concentrations of fatty acids vary hugely depending on individuals and body sites (e.g.: Abdelmagid et al., 2015; doi: 10.1371/journal.pone.0116195).

We have repeated an esterification assay with 30 μ M instead of the initial 300 μ M and show that the Lip2 esterification activities remained apparent (Fig. S5A).

We agree with this reviewer about the artificial nature of our *in vivo* study. This is now acknowledged and discussed in the manuscript (Lines 452-454). Further studies are needed to characterize the full extent of staphylococcal lipase-mediated changes of the host lipid landscape.

Additional comments:

1. The medium used in all the growth curve and survival studies are not chemically defined media. Although it seems that wt *S. aureus* and lip mutants have no growth differences when no PA and cholesterol are introduced, these rich media could introduce artifacts due to the rich nutrients and growth factors within the media. Have the authors considered testing the growth conditions with PA and cholesterol in chemically defined medium and see if Lip2 is truly able to rescue *S. aureus* from AFA via esterification with cholesterol?

We thank this reviewer for this comment about using a chemically defined medium (CDM). Using a CDM (Pohl et al., 2009; doi: 10.1128/jb.01492-08), we now show that WT *S. aureus* and the lipase-deficient mutant are similarly killed by 50 μ M PA. However, only WT is protected by cholesterol (Fig. S1D).

2. The dose of both cholesterol and linoleic acid used in the *in vitro* enzymatic experiment in Fig. 5 are relatively high (300 μ M). Under this single-dose condition, it is difficult to determine whether the esterification from Lip2 is truly due to the substrate-

specific catalytic activity of purified Lip2 enzyme, or this esterification was catalyzed by Lip2 because both the LA and cholesterol concentration was set to a high. The authors should consider doing additional titration experiments to demonstrate the catalytic activities from Lip2 is truly dose-dependent and specific to both substrates. Also, have the authors tested the possible esterification activities of purified Lip1?

Linoleic acid (LA) represents one of the most abundant fatty acids in the human body, reaching up to 5 mM in the blood (Abdelmagid et al., 2015; doi: 10.1371/journal.pone.0116195). Therefore, we did not initially consider 300 μ M to be especially high. We used 300 μ M for the experiment presented in Fig. 5A because this concentration allows for robust lipid detection via TLC. Similar studies have even used 400 μ M (Ping Teoh et al., 2021, DOI: 10.1073/pnas.2022720118). To test the specificity of the Lip2-mediated esterification, we have repeated the experiment (Fig. 5A) with ten times less lipids (30 μ M). With this concentration, Lip2 activity over time is still apparent (Fig. S5A). Using increasing concentration of LA in such an assay, we also show that Lip2-mediated cholesteryl ester production is dose-dependent (Fig. S5B).

We have indeed extensively tested the purified Lip1 to characterize novel activities. We did not observe any Lip1-mediated esterification of cholesterol. We opted against including these data in the current manuscript as (i) the Lip1 data are part of the follow-up study further characterizing additional staphylococcal lipases, and (ii) we believe to provide sufficient data showing that Lip2 rather than Lip1 is required for cholesterol esterification (Figs 3 to 5 and Fig. S6). We beg the esteemed referee's indulgence.

3. On Fig. 5A., as the time proceeded to 120 min, the cholesterol concentration stopped decreasing around 20-25 μ M. Is this because of the loss of Lip2's enzymatic activity

after 2 hours? Or a minimum concentration of 20-25 μM cholesterol is essential for the Lip2 to esterify the AFA?

The new data of Fig. S5A suggests that Lip2 can be active below cholesterol concentrations of 20-25 μM . Fig. 5A shows that after 120 min an equilibrium state between Lip2-mediated ester production and lipolysis is probably reached, given the known capacity of Lip2 to hydrolyse cholesteryl esters (Hines et al., 2020, doi: 10.1128/mSphere.00339-20).

4. The *in vivo* data is modest and artificial. Why mice needed supplementation of cholesterol and sapienic acid? Do mice produce antimicrobial fatty acid?

Mice do produce antimicrobial fatty acids, including palmitoleic acid (C16:1) and oleic acid (C18:1), which contain one *cis*-9 double bond (Georgel et al., 2005; doi:10.1128/IAI.73.8.4512–4521.2005). *S. aureus* can detoxify palmitoleic acid and oleic acid by utilizing OhyA (oleate hydratase) to catalyse the hydration of *cis*-9 double bonds Subramanian et al., 2019; doi: 10.1074/jbc.RA119.008439). Importantly, sapienic acid (C16:1 with a *cis*-6 double bond) cannot be detoxified by OhyA, but by Lip2 (Fig. S5C). Therefore, we selected sapienic acid for *in vivo* studies with a mouse skin colonization model of atopic dermatitis. In a different study with this model, we did not observe any differences in colonization between wild-type *S. aureus* JE2 and the lipase-deficient mutant (Nguyen et al., 2017; doi: 10.1016/j.ijmm.2017.11.013). We reasoned that extensive tape-stripping strongly depletes lipids (lines 388 - 390), which may at least partly explain why tape-stripping of mouse skin significantly increases colonization by *S. aureus* (Focken et al., 2023; 10.1016/j.celrep.2023.113148). Therefore, we developed a “humanized” mouse model where sapienic acid and

cholesterol were supplemented. We agree with this reviewer that our model is artificial (Lines 452-454). Nonetheless, albeit modest, a clear difference was visible for WT *S. aureus* after only 24 hours and a local application of lipids. We anticipate that this difference could increase after a few days (e.g.: (Georgel et al., 2005; doi:10.1128/IAI.73.8.4512–4521.2005) or could be vital within the context of a competition with lipase-deficient bacteria in an AFA-rich environment.

5. From Fig. 1 to Fig. 4, the authors tested the growth advantage of wt *S. aureus* growing in medium supplemented with PA or PA + cholesterol combination. Besides the 50uM inhibition concentration of PA mentioned in line 97 of the manuscript, the rest of the experiment in Fig. 1 to Fig. 4 do not clearly listed the actual concentration of PA and cholesterol used in the main text and figure caption. (However, the concentration of PA and cholesterol was both clearly listed on the SI section.) The main text should also have all the concentration clearly listed. Also, it is important to provide additional rationales of how the PA, LA and cholesterol concentration used are biologically relevant.

We thank the reviewers for this comment. Concentrations are now indicated in all figure legends. The biological relevance of the concentrations used has been discussed in detail at the beginning of this response letter.

Reviewer #2 (Remarks to the Author):

In this manuscript, Tchoupa et al study the role of two *Staphylococcus aureus* secreted lipases in the resistance to antimicrobial fatty acids (AFAs). The authors conduct compelling and well-controlled in vitro studies to demonstrate that AFAs are detoxified

when *S. aureus* is grown in the presence of cholesterol. AFA detoxification in the presence of cholesterol was dependent on the Lipase Lip2 (also known as Geh or Sal2), but not Lip1 (Sal1). Lip2 is a well-characterized glycerol ester hydrolase that can hydrolyze ester linked fatty acids from triglycerides, but also lipoprotein, phospholipids, and cholesterol esters. Several lipases of this class are known to also exhibit reversible esterification activity, whereby they form fatty acid esters with cholesterol or other alcohols (FAME activity). *S. aureus* is known to produce an enzyme with FAME activity, but the enzyme responsible has remained elusive. Because Lip2 was able to detoxify antimicrobial fatty acids in the presence of cholesterol in a Lip2 dependent manner, the authors reasoned this might be due to FAME activity of Lip2 leading to the formation of less toxic cholesterol esters. The authors use in vitro biochemical assays along with TLC and Mass spectrometry to show that this is true. They then follow-up with a series of studies to ascertain how the presence of cholesterol and lip2 might be preventing toxic effects on bacterial cells and find that cholesterol and lipid binding to the cell surface is unchanged and that membrane potential is lost regardless of lipase presence. Lastly, the authors survey several thousand *S. aureus* isolates and find that Lip2 is well conserved with relatively low levels of prophage insertional disruption. Although low, there was a difference in the number of strains with prophage insertion from the blood/nose (~2%) versus the skin (~1%), which might indicate a role in skin colonization. The authors try to conduct an experiment to get at a role for Lip2 in promoting skin colonization and find that addition of sapienic acid and cholesterol along with *S. aureus* to the skin seems to promote survival of lip2 containing strains within the skin tissue.

Overall, this study is well-controlled and clearly demonstrates that Lip2 has FAME activity and promotes survival in the presence of antimicrobial fatty acids when grown

with cholesterol in vitro (Figures 1-5). These figures represent a major strength of the study. However, the conclusions related to figures 6 and 7 are not felt to be fully justified at this time. In addition, the discussion does not fully place this work within the context of the literature on known lipases with FAME activity or provide a compelling synthesis on the potential for the reversible activity of Lip2 (e.g. when might hydrolysis predominant over esterification etc and would would the advantage be based on recent studies in the literature).

We thank this reviewer for the thorough analysis of our manuscript. We have now amended Figure 6 and the accompanying text for clarity. The conclusions related to figure 7 have been changed in accordance with the data. The discussion has been expanded to elaborate on the reversible activity of Lip2 and other bacterial lipases.

1. Figure 6 - Using cholesterol and fatty acid binding studies along with disruption of membrane potential the authors reason that cholesterol ester formation is the only plausible means of protection from AFA in the presence cholesterol. However, it is not clear how this is happening, especially if membrane damage (loss of membrane potential) still occurs in all conditions? How exactly are AFA cholesterol esters protective? This could be more clearly described or considered in the discussion.

We agree with this reviewer that this section of our manuscript may seem counterintuitive. We have amended this section for clarity (lines 298 - 300, and lines 344 - 349). As also shown for other AFA-detoxifying enzymes (Subramanian et al., 2019; doi: 10.1074/jbc.RA119.008439), Lip2 does not prevent the initial "impact" caused by toxic concentrations of PA even in the presence of cholesterol, as demonstrated with fatty acid studies (Fig. 6B) and membrane potential assessment

upon short exposure (30 min) to AFAs (Fig. 6C). We now provide additional data revealing that Lip2-expressing bacteria, which were given enough time (24 h) to utilize cholesterol to detoxify AFAs into innocuous cholesteryl esters, displayed a membrane potential resembling that of the untreated controls (Fig. 6D).

2. Although the difference in the percentage of strains (2.1% in blood versus 0.7% on skin) containing a phage insertion in lip2 is noted by the authors it is not clear how well this data justifies the possibility that lip2 is required for skin colonization over an equally important requirement at other sites. 99% of skin isolates do not have a disruption in lip2 versus 98% of blood and nose. It is not clear that there is a notable correlation with phage insertion that is supported by this data, except that the majority of *S. aureus* isolates do not have phage disruption regardless of site. This data might argue that the lipase is important for host microbe interactions overall since most strains lack a phage disruption, but it is not clear it says much about skin over other sites. Can the authors either justify this claim better or modify their conclusions to better reflect the results.

We thank the reviewer for this valid concern. We have now changed our abstract and the conclusion of our genomic analysis in line with our data (e.g.: “These results suggest that an intact lip2 may be required for successful colonization and/or infection by *S. aureus*”).

3. Figure 7 infection data do not really test the hypothesis the lipase mediated esterification is beneficial for *S. aureus* colonization of the skin. The experiment supplies sapienic acid and cholesterol together with *S. aureus* rather than testing the role of the lipid environment of the skin itself. As it stands there is no strong evidence

that Lip2 promotes skin colonization via esterification of cholesterol to alleviate AFA stress. Conclusions from this experiment should be calibrated to reflect the strengths and weaknesses of the experimental design. This should be addressed in the discussion.

We have amended our conclusion to reflect the weaknesses of our experimental design, which we have also discussed (lines 443 - 459).

4. Reference 1 of the manuscript has previously shown that cholesterol esters themselves are substrates for ester hydrolysis by Lip2 in vitro. The ability to hydrolyze FA esters versus esterification of free FA is heavily influenced by conditions, lipid/FA, and alcohol abundance, and other conditions, especially in vitro. The authors might consider taking some time to reconcile their results with the hydrolysis activity shown in Ref 1 and provide their take on the potential for the reversible activity of Lip2 (e.g. when might hydrolysis predominant over esterification). For example, there is an interesting paper from the Orth lab (which is already referenced in the work for methodological reasons, see 56) that highlights some of the potential diverse functions for lipase FAME activity in pathogenesis. There seems to be a lot of opportunity to put the findings of this work in a broader context and reflect on the importance of the potential of this shift from hydrolysis to esterification.

We thank this reviewer for this suggestion. We have now discussed in great detail what might tip the balance for Lip2 activities towards hydrolysis or esterification (lines 460 - 475, and 492 - 513). We also elaborate on FAME activity in the broader context of pathogenesis (lines 460 -475, and 492 - 513).

Reviewer #3 (Remarks to the Author):

This manuscript concludes that the Lip2 lipase of *S. aureus* has a reverse role in esterifying fatty acids to cholesterol, and that this function has a critical role in detoxifying antimicrobial fatty acids encountered on skin and nasal secretions. The authors provide a convincing demonstration that Lip2 accounts for the fatty acid modifying enzyme FAME activity that has been known of for many years but never identified. The biochemical data and growth analyses are supportive of a role for Lip2 in detoxifying antimicrobial fatty acids through esterification to cholesterol. The biochemical data are strong and leaning toward excessive, but conclusions of relevance to skin colonization need to be tempered.

We thank this referee for positively reviewing our work. We have now changed the conclusion of our skin colonisation model to better reflect the data.

Major comments:

1. Data in support of lipase esterification of free fatty acid to cholesterol is convincing, as are the in vitro data indicating that toxicity of palmitoleic acid is ameliorated in the presence of cholesterol. However, on human skin, sebaceous secretions contain cholesterol, free fatty acids and triglyceride, of which the latter is most likely a preferred substrate for lipase. A question that should be addressed is whether lipase will promote esterification of free fatty acids to cholesterol in the presence of triglyceride, which is also abundant in sebaceous secretions.

We have performed growth assay in rich media (nutrient broth, basic medium, and tryptic soy broth), which arguably all contain some triglycerides derived from meat extract, yeast extract, or soya bean, respectively. In all these media, we could observe

the Lip2-dependent protective effects of cholesterol against antimicrobial fatty acids (e.g.: Figs 1-4). Moreover, bacteria-conditioned media (TSB) used to demonstrate that Lip2 released by *S. aureus* can esterify cholesterol with AFAs (Figs. 5D and S6) probably contain some triglycerides too. Since a recent study from the Xu lab (Pruitt et al., 2024; doi: 10.1128/msphere.00368) made the following statement about Lip2/Geh: "...glycerol ester hydrolase (Geh) is the primary lipase hydrolyzing cholesteryl esters and, to a lesser extent, triglycerides...", it is not certain that triglycerides are "preferred substrate[s] for lipase". However, we have performed an additional experiment with the purified Lip2, cholesterol, palmitoleic acid, and its triacyl glycerol ester, glyceryl tripalmitoleate, and we now show that Lip2 promotes the esterification of free fatty acids to cholesterol even in the presence of glyceryl tripalmitoleate (Fig. S5D). Accordingly, glyceryl tripalmitoleate could not abrogate cholesterol protective effects against palmitoleic acid (Fig. S1E).

2. Skin is known to have acidic pH. Have the authors assessed the influence of pH on the esterification reaction? Have previous studies on FAME activity addressed the optimum pH for this reaction?

We thank this reviewer for these important questions. The answers are now included in the manuscript (lines 460 - 475). Briefly, previous studies have determined that the optimum FAME activity occurred between pH 5.5 and pH 6 (Mortensen et al., 1992; doi: 10.1099/00222615-36-4-293). We could confirm this during our preliminary experiments with recombinant Lip2, where acidifying the buffer to pH 5 resulted in a strong decrease in the FAME activity. Since this had been previously demonstrated, we only included data obtained with the optimum pH (pH 6) in the first version of our manuscript. We are now providing data showing that the FAME activity is all but

abrogated at pH8 (Fig. S5D), the previously reported optimum pH for lipase (hydrolysis) activity (Cadieux et al., 2014; doi: 10.1128/JB.02044-14.). Thus, the acidic pH of the skin can limit FAME activity if below 5.5, while a pH between 5.5 and 6 promotes FAME activity and is associated with heightened colonisation rates by *S. aureus* in atopic dermatitis patients (Hülpüsch et. al., 2020; doi: 10.1111/all.14461).

3. Lines 260-279 describing how cholesterol does not prevent membrane-damaging effects of antimicrobial fatty acids is not clearly written. First the authors propose a potential but arguably also imaginative alternative explanation for how cholesterol might alleviate toxicity of fatty acids; namely that a secreted lipase might promote microbial binding to cholesterol such that the resulting aggregates are less susceptible to fatty acid. The authors conduct an experiment that negates this hypothesis, and then use a click chemistry experiment to suggest that cholesterol does not prevent membrane targeting of AFA's. This conclusion seems counterintuitive since cholesterol alleviates toxicity and toxicity is attributed to membrane damage. Note that the click chemistry experiment SHOULD demonstrate a significant difference in the distribution of free fatty acid, which can either be incorporated into phospholipid, esterified to cholesterol, or accumulate in the membrane as a free fatty acid, which is what promotes toxicity. As such, this entire section fails to make a strong point and detracts from the linear narrative of the story.

We have now amended the text for clarity (lines 298 - 300). The Torres Lab has recently demonstrated that Geh/Lip2 was not only secreted into the extracellular milieu, but also associated with the bacterial cell envelope (Zheng et al., 2021, doi: 10.1038/s41467-021-26517-z). Therefore, surface-bound Lip2 is not only the mere product of our imagination. We see it as a means for lipase-producing bacteria to make sure that they

fully benefit from lipase activities. We are not sure whether Lip2 can use the fatty acid analogue (palmitoleic acid alkyne) for esterification with cholesterol. If so, the timeframe used for this assay (~ 1 hour) may not be sufficient to observe a significant difference in the distribution of the free fatty acid analogue. In line with our growth assays and as discussed in the response to this reviewer's minor comment 1, we think that neither Lip2 nor cholesterol prevents the initial interaction of AFAs with *S. aureus* membrane (Lines 311 - 316). The ensuing damage does not necessarily result in bacterial cell death. We now provide evidence that Lip2-proficient bacteria, which exploited cholesterol to survive AFA exposure presented no membrane damages (Fig. 6D). We do understand that this Fig. 6 adds a layer of complexity to our story, which could be deemed as distracting, but we believe that the data support a model whereby bacteria have to sense, respond to, and overcome a challenge.

4. Lines 336 to 349 comprise a section entitled "The capacity to manipulate cholesterol governs skin colonization by *S. aureus*". Contrary to the certainty implied by the title of this section, the text of the results section is less emphatic, indicating that the lipase deficient mutant "was largely" unaffected by cholesterol application, while the Lip-2 proficient wild type "appeared to benefit" from cholesterol application, and the data "strongly suggest" that *S. aureus* utilizes lipase to manipulate environmental lipids. The authors are correct to imply some uncertainty as to the rigor of this conclusion.

We have changed the title of this section to "Cholesterol contributes to the protection of *S. aureus* from AFAs on the skin" to add some uncertainty in accordance with the modest data and the limitations of the model used.

5. Regarding the colonization model, it is recognized that it is challenging to select an appropriate model. The model that is chosen can be useful but is subject to limitations; namely that it mimics atopic dermatitis where there is a deficiency in the skin barrier, and indeed the authors acknowledge this by stating that the taping technique strips the skin of its lipids, necessitating that the lipid barrier needs to be reconstituted by addition of cholesterol and free fatty acids in order to observe a phenotype between wild type and Lip2 deficient mutant. There are a lot of caveats here. One thought to consider here is that the same taping technique that has previously been used to recover and define the lipid composition of sebaceous secretions could also be applied post-challenge to determine if there is evidence of sapienic acid esterification to cholesterol on the skin of mice. If the authors apply a cholesterol of known composition that differs from the predominant cholesterol in the skin of mice, it should be feasible to determine if it is being modified by esterification of sapienic acid, which is not produced by mice.

We acknowledge that deciphering the impact of colonisation by *S. aureus* on the skin lipid landscape would be required to demonstrate that host-derived fatty acids are indeed esterified in a Lip2-dependent manner. However, this will require optimisation at all steps (sample collection, lipid extraction and lipidomics). Instead of embarking on such a long, uncertain though exciting journey, we have changed our conclusions to better reflect the available data.

6. The authors should acknowledge that Lip2 is equivalent to the glycerol ester hydrolase Geh, which was also shown to modulate the immune response by deacylating *S. aureus* lipoproteins, such that the host innate immune response mediated by TLR2 is dampened. (PNAS 2019; 116:3764-3773). While it is difficult to relate the authors current data on “colonization” to previous work on lip2 deficiency and

persistence in organs, it should at least be acknowledged that the lip2/geh deficiency could alter the cytokine response to infection. Notably, the authors did not observe any impact of lip2 and cholesterol on what is presumed to be surface adherent bacteria, whereas the minced tissue samples presumably representing tissue invasion indicated significantly more bacteria in the lip2 deficient strain. Some H&E staining of the infected tissue and pathology analysis would be informative here.

We do acknowledge that Lip2 is “also referred to as Geh or Sal2” (line 76). We had cited that paper from the Alonso lab in the previous version of our manuscript as one of the few studies to have demonstrated that Geh/Lip2 is a virulence factor (lines 435 - 436). We are now discussing the immunomodulatory roles of Geh/Lip2 (lines 506 - 513). The Alonso lab has nicely demonstrated that Geh blunts the pro-inflammatory response provoked by *S. aureus in vivo*, which results in delayed microbial clearance (Chen et al., 2019; 10.1073/pnas.1817248116). As we cannot exclude an altered cytokine response to our lipase-deficient strain compared to WT, we have restricted our conclusions to the effects of the lipid supplementation on each strain, although we could not see any statistically significant difference in “tissue invasion” (Fig. 7D). However, the seemingly improved ability of Δ lip to invade tissue upon sapienic acid treatment, i.e., higher CFU for Δ lip compared to WT, cannot be explained by an increased proinflammatory potential, which the Alonzo lab has shown to lead to decreased CFU and rapid clearance of the infection in general. Besides, the Alonzo lab has recently discovered that the incorporation of unsaturated fatty acids into lipoproteins greatly hampers the immunomodulatory role of Geh/Lip2. This scenario is plausible in our model where bacteria were treated with an unsaturated fatty acid (sapienic acid) during skin colonization. We think that this difference in CFU could simply reflect the fact that Δ lip is more “resistant” to antimicrobial fatty than WT

because the mutant is defective in releasing more toxic fatty acids from its environment as demonstrated by McGavin and Heinrichs labs (Cadieux et al., 2014; doi: 10.1128/JB.02044-14). However, we did not elaborate on this in our discussion because the CFU data are not statistically significant.

Although H&E staining of the infected tissue and pathology analysis would give us more clues as to what is happening *in vivo*, we consider these experiments to be beyond the scope of the current study about bacterial colonization.

Minor comments:

1. Even in the presence of cholesterol, wild type bacteria still exhibited an ~ 7.5h lag phase when cultured with palmitoleic acid compared to media alone. Consequently, it appears that palmitoleic acid does retain some toxicity in presence of cholesterol.

Our data indicate that the presence of cholesterol does not prevent palmitoleic acid (PA) from binding to bacteria (Fig. 6B) and compromise the integrity of bacterial membrane (Fig. 6C). Therefore, bacteria must recover from the initial “impact” of PA treatment before resuming growth after PA detoxification.

2. The authors use overnight (and presumably) stationary phase cultures as inoculum for the growth assays. There is no mention here of washing the cells prior to inoculating plates for growth assays. Does the 7.5h lag relate to the ability of lipase carried over in the inoculum to esterify fatty acid to cholesterol? Would the lag phase be longer if the bacteria were washed prior to inoculation? Note that for the infection experiments, the methodology specifies that the bacteria were washed prior to challenge.

We used indeed overnight and stationary phase cultures as inoculum for growth assays. Generally, we do not wash bacteria prior to inoculating plates for growth assays. When bacterial cultures are centrifuged and the spent media discarded, so are membrane vesicles, dead cells, protein aggregates that can sequester fatty acids and prevent them from reaching live cells (Kengmo Tchoupa et al., *msphere*, 2020; doi: [10.1128/msphere.00804-20](https://doi.org/10.1128/msphere.00804-20)). Therefore, washed bacteria display an increased susceptibility to AFAs. In such a context, the lag phase in the presence of inhibitory concentrations of PA and cholesterol was too long (over 12 hours) for growth curves to be consistent in nutrient broth or basic medium. However, using TSB, we were able to demonstrate that lipases produced during the growth assay (i.e., with bacteria washed prior to inoculation), utilized cholesterol to protect WT *S. aureus* from palmitoleic acid (Fig. S10).

3. Exponential phase cells are more susceptible to the bactericidal activity of antimicrobial fatty acids compared to stationary phase cells. How would the data look if exponential phase cells were used as inoculum?

In *Staphylococcus aureus*, lipases are under the control of the quorum-sensing system Agr. Arguably, bacteria in the exponential phase are not producing a lot of lipases. Therefore, we were not surprised to see that bacteria grown to an optical density of 2.5 (after 3 hours) could not fully benefit from cholesterol to resist LA (Fig. S2C).

4. As the lead figure in the paper, the choice of symbols and colors for Fig 1 is very poor, making it difficult to follow.

Symbols and colours have been changed in Fig. 1 for clarity.

5. From Fig's 1A and 1D it is apparent that WT *S. aureus* exhibits a significantly lower growth rate in media supplemented with palmitoleic acid and cholesterol compared to medium alone, and also reaches a significantly lower final OD₆₀₀. Consequently, it appears that there is still significant toxicity associated with the palmitoleic acid. Also, expressing data as "Area Under the Curve" does not show to what extent there is an extended lag phase, which is also a sign of stress.

An extended lag phase and a significantly lower final OD₆₀₀ actually result in lower "Area Under the Curve", which we showed in Figs. 1 to 5. Changes in optical density are not only due to live bacteria, but secreted proteins and membrane vesicles. The secretion of proteins can be hugely impacted by antimicrobial fatty acids (Arsic et al., 2012; doi: 10.1371/journal.pone.0045952). Therefore, we looked at colony forming units (Fig. 1C) and bacterial spots (Figure 3 A-D) to confirm that lipase-expressing bacteria were able to grow in media supplemented with antimicrobial fatty acids and cholesterol. The increased lag phase in the presence of AFAs reflects the time needed for enzymatic detoxification (see also Subramanian et al., 2019; doi: 10.1074/jbc.RA119.008439) and probably the fact that some bacterial cells do not recover from the initial damage by AFAs.

REVIEWERS' COMMENTS:

Reviewer #1 (Remarks to the Author):

The authors have addressed all my prior concerns and the revised manuscript is more balanced and accurate. All in all, the authors are to be commended for addressing the reviewers critiques.

Reviewer #2 (Remarks to the Author):

The authors have addressed my initial concerns with new data, modified conclusions, and additional depth in the discussion. The new data provided addresses several points of confusion and strengthens the manuscript. Overall, this is a well-conducted study with interesting implications for the physiological roles of lipases during infection.

Reviewer #3 (Remarks to the Author):

This revised manuscript solves a long standing mystery in research on secreted enzymes of *S. aureus*; namely the identification of glycerol ester hydrolase/SAL2 as the fatty acid modifying enzyme FAME responsible for detoxification of unsaturated fatty acids through esterification to cholesterol. This is strongly supported by biochemical data, and demonstration that the acidic pH optimum of the esterification reaction is supportive of function on skin. The *in vivo* phenotype is modest, but is again supportive of lipase promoting skin colonization and invasion through esterification of toxic fatty acids to cholesterol. The authors have conducted additional experimentation to address comments of reviewers 1 and 2, and have tempered conclusions on the impact of the *in vivo* data, while adding more extensive discussion to better place the work in context of previous findings on lipase functions and relation to FAME activity.